# Postpartum breast cancer has a distinct molecular profile that predicts poor outcomes

Sonali Jindal [1,2,9], Nathan D. Pennock [1,9], Duanchen Sun[3,9], Wesley Horton[1,3], Michelle K. Ozaki [1], Jayasri Narasimhan[1], Alexandra Q. Bartlett[1], Sheila Weinmann[4], Paul E. Goss[5], Virginia F. Borges[6,7], Zheng Xia [2,3,8,10] & Pepper Schedin [1,2,7,10✉]

Young women's breast cancer (YWBC) has poor prognosis and known interactions with parity. Women diagnosed within 5–10 years of childbirth, defined as postpartum breast cancer (PPBC), have poorer prognosis compared to age, stage, and biologic subtype-matched nulliparous patients. Genomic differences that explain this poor prognosis remain unknown. In this study, using RNA expression data from clinically matched estrogen receptor positive (ER+) cases ($n = 16$), we observe that ER+ YWBC can be differentiated based on a postpartum or nulliparous diagnosis. The gene expression signatures of PPBC are consistent with increased cell cycle, T-cell activation and reduced estrogen receptor and TP53 signaling. When applied to a large YWBC cohort, these signatures for ER+ PPBC associate with significantly reduced 15-year survival rates in high compared to low expressing cases. Cumulatively these results provide evidence that PPBC is a unique entity within YWBC with poor prognostic phenotypes.

[1] Department of Cell, Developmental and Cancer Biology, Oregon Health & Science University, Portland, OR, USA. [2] Knight Cancer Institute, Oregon Health & Science University, Portland, OR, USA. [3] Computational Biology Program, Oregon Health & Science University, Portland, OR, USA. [4] Center for Health Research, Kaiser Permanente Northwest, 3800N. Interstate Ave., Portland, OR, USA. [5] Massachusetts General Hospital Cancer Center, Harvard University, Boston, MA, USA. [6] Division of Medical Oncology, Department of Medicine, University of Colorado Anschutz Medical Campus, Aurora, CO, USA. [7] Young Women's Breast Cancer Translational Program, University of Colorado Cancer Center, Aurora, CO, USA. [8] Department of Molecular Microbiology and Immunology, Oregon Health & Science University, Portland, OR, USA. [9] These authors contributed equally: Sonali Jindal, Nathan D. Pennock, Duanchen Sun. [10] These authors jointly supervised this work: Zheng Xia, Pepper Schedin. ✉email: schedin@ohsu.edu

 1

Breast cancer incidence is bimodal, with peaks ~45 and 65 years of age referred to as early and late-onset disease, respectively[1–5]. As breast cancer risk does not increase linearly with age, it is suggested that early and late-onset breast cancer are distinct entities with their own risk factors and molecular signatures[2,4]. Early-onset breast cancer, also known as young women's breast cancer (YWBC), is a global concern. YWBC accounts for ~11% of all new breast cancer diagnoses in the United States[6–8], and the incidence of YWBC in many developing countries is higher[9,10]. Further, the incidence of YWBC is increasing world-wide[11–14]. A recent retrospective SEER registry study representing 25% of the US population reported a 1.62 (1.16–2.09) fold increase in the incidence of YWBC between 2000 and 2015 alone, with increased incidence across all races and ethnicities[13]. In addition, compared with late-onset breast cancers, YWBC is enriched in poor prognostic tumor features[15–18], has high levels of mortality[15,18–21], and has experienced limited gains in treatment efficacy[16,22]. Thus, an improved understanding of the underpinnings of YWBC is needed to effectively combat this poor prognostic disease.

An elevated proportion of poor prognostic hormone receptor (HR)-negative and HER2-positive breast cancers is often cited to account for the adverse outcomes in young patients[15–18]. However, several lines of evidence suggest that differences in intrinsic biologic subtypes—including estrogen receptor (ER) and HER2 status—do not wholly account for the observed increased mortality. For example, in the same US SEER study reporting a 1.62-fold increase in YWBC since 2000, the increase in incidence was attributed exclusively to ER-positive (ER+) disease[13]. Further, contrary to expectations that luminal A and B breast cancers are less deadly in young women, a National Comprehensive Cancer Network study of 17,575 women with stage I–III breast cancer reports higher breast cancer mortality in young women with luminal A (HR 2.1; 95% CI, 1.4–3.2) and B (HR 1.4; 95% CI, 1.1–1.9) cancers compared with young women with triple-negative or HER2+ cancers[23]. Similar trends have also been reported in young Chinese women[24]. These studies provide further rationale to explore early-onset breast cancers as distinct entities whose biology is not fully explained by differing ER or HER2 status.

Since breast cancer incidence is influenced by parity[25–30], one possible explanation for the poor prognosis in young patients is that cancer outcomes are associated with childbirth. A recent meta-analysis of 41 studies addressed whether YWBC outcomes are differentially influenced by a diagnosis during pregnancy or the postpartum period. This analysis revealed a higher risk of death only in women diagnosed postpartum (HR 1.79; 95 % CI 1.39–2.29)[31]. Further, these and other studies found that a diagnosis within 5–10 years of a recent pregnancy, referred to as postpartum breast cancer (PPBC)[32], independently associated with a two- to threefold increased risk of death in both ER+ and ER− disease[33,34]. Conversely, studies find that a diagnosis during pregnancy is not associated with poorer outcomes[35–37]. Combined, these studies implicate the existence of a postpartum event that negatively impacts breast cancer prognosis. In women, the postpartum window coincides with a developmental process known as weaning-induced breast involution, a process demonstrated to promote breast cancer development and metastasis in rodent models[38–41]. Given that ~50% of all YWBC are diagnosed within 10 years of a completed pregnancy[33,34], further investigation into the impact of postpartum breast involution on tumor biology is warranted.

Involution is a physiologically normal process that remodels the epithelial-dense, lactational gland to a pre-pregnant-like, non-secretory state[42–44]. In female rodents, where the involution process has been extensively studied, >80% of the lactational mammary epithelium dies as part of a developmentally regulated tissue remodeling process[42–44]. This process coordinates responses of mucosal immunity, fibroblast activation, lymphangiogenesis, and wound-like extracellular matrix deposition[45–49]. In addition to involution creating a transient stromal microenvironment favorable for the expansion and spread of primary tumor cells, involution also durably alters murine mammary tumors. This is evidenced by features of elevated COX-2 expression, increased lymphangiogenesis-inducing capability, augmentation of a tumor-associated immune milieu, and enhanced tumor growth and dissemination phenotypes, all of which persist beyond the period of weaning-induced gland involution in rodents[38,47,50]. Collectively, preclinical studies of PPBC suggest that YWBC may be durably influenced by the transitory developmental processes of mammary gland involution, which may result in distinct gene expression profiles predictive of poor outcomes.

Here, we address whether YWBC can be delineated into distinct molecular subtypes based on a nulliparous or postpartum diagnosis. We focus on ER+ disease as an under-investigated breast cancer subtype accounting for more deaths overall than ER− disease[34,51,52]. We perform comparative RNA Seq expression analyses on treatment-naive formalin-fixed, paraffin-embedded (FFPE) breast cancer tissues from young patients using tumor stage-matched, ER+ postpartum (PPBC), and nulliparous breast cancers (NPBC). We validate gene expression results using multiplex immunohistochemistry (mIHC). We find that PPBC associates with enhanced signatures of cell cycle control, T-cell activation and exhaustion, decreased ER signaling, and altered P53 signaling compared with matched cases diagnosed in nulliparous women. This study strongly supports the hypothesis that normal postpartum breast involution durably alters breast cancer intrinsic and extrinsic factors predictive of disease progression.

## Results

**PPBC RNA expression profile is distinct from NPBC.** To gain insight into the features that could lead to poorer outcomes in PPBC patients, we focused our analyses on clinically determined ER+ cases, as ≥65% of all young breast cancer patients (≤45 years of age) are diagnosed with ER+ disease[53]. Further, young women's ER+ breast cancers have threefold increased likelihood of progressing to metastatic disease when diagnosed postpartum (PPBC) compared with nulliparous cases (Nulliparous Breast Cancer–NPBC)[34]. To obtain a cohort of age and stage-matched, treatment-naive, ER+ NPBC and PPBC cases, we performed chart review for patient age, pregnancy history, tumor stage, subtype, and treatment history. Of 40 selected cases, 16 ER+ cases (PPBC $n = 9$, NPBC $n = 7$) yielded RNA in sufficient quantity and quality to advance to RNA sequencing and subsequent gene expression analyses. Unsupervised hierarchical clustering of these 16 samples across all 14,830 expressed genes yielded separation of 14 of the 16 samples based on parity status (Fig. 1a, nulliparous (blue) vs postpartum (black)). Of note, these cases did not separate based on clinical stage, suggesting parity history is more predictive of tumor gene expression than tumor clinical stage in this young cohort. We identified the most differentially regulated genes between NPBC and PPBC specimens utilizing DESeq2 bioinformatics program and found 364 genes with a false discovery rate (FDR) of ≤0.1 (adjusted $p$ value). Unsupervised clustering of these 364 genes resulted in only one misalignment between the two parity groups (Fig. 1b). To determine whether these differentially expressed genes represent a coordinated change in tumor biology, we used STRING[54] database analysis, which predicts protein–protein interactions across a

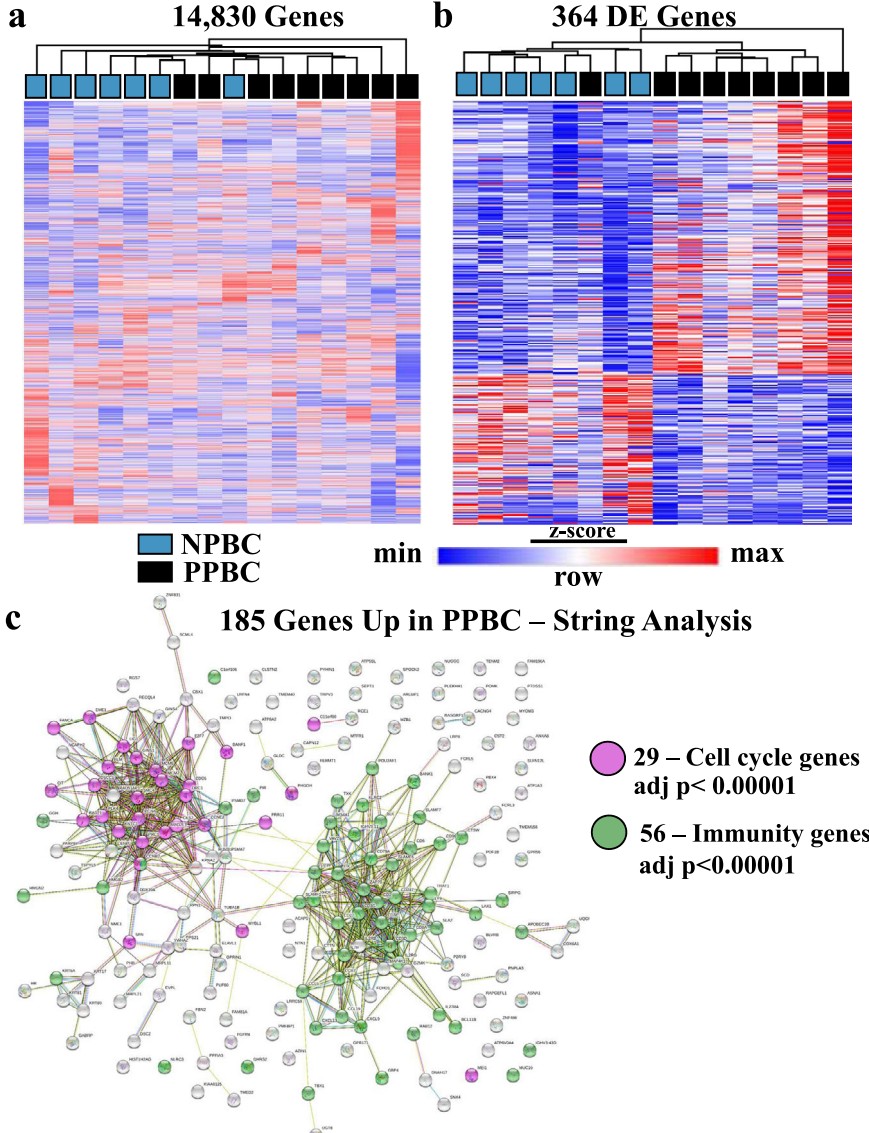

**Fig. 1 RNA expression profiling separates postpartum breast cancer (PPBC) from nulliparous breast cancer (NPBC).** RNA seq, performed on RNA obtained from FFPE specimens of primary ER+ breast cancer from patients 45 years of age or younger, reveals parity effect. Clustering analysis derived from RNA expression profiles of biologically independent samples of nulliparous breast cancer (NPBC, blue, $n = 7$) and postpartum breast cancer (PPBC, black, $n = 9$). **a** Euclidean hierarchical clustering of the 14,830 genes determined to be expressed above background. **b** Euclidean hierarchical clustering based upon 364 differentially expressed genes between PPBC and NPBC determined by DESeq2 with an FDR < 0.1. **c** STRING database clustering analysis[54] of 185 upregulated PPBC genes generates two distinct biological clusters of statistical significance (adj = adjusted. *p* values adjusted according to Benjamini–Hochberg for multiple comparisons).

variety of annotated "omics studies". We identified two dominant (*p* value < 0.00001) clusters of genes that increased in PPBC compared with NPBC. One of these clusters is associated with cell cycle programs (Fig. 1c, purple) and the other with immunity (Fig. 1c, green).

**Gene set enrichment characteristics of PPBC.** We next performed rank-based gene set enrichment analysis (GSEA) on PPBC compared with NPBC. We observed enrichment in pathways associated with six distinct biological processes in PPBC compared with NPBC (Fig. 2a–f, Supplementary Data 1). Consistent with the STRING analysis (Fig. 1c), we observed enrichment for cell cycle and proliferation signatures (Fig. 2a), as well as signatures associated with cell death and DNA repair (Fig. 2b). We also observed enrichment in the T-cell presence-activation signature (Fig. 2c), an observation that provides cell-specific

insight into the enriched immunity signature detected by STRING analyses. Surprisingly, even though all cases were determined to be definitively ER+ (Supplementary Table 1a), in the PPBC cohort we observed enrichment of gene expression profiles associated with ER-negative breast cancers[55] (Fig. 2d). Further, in PPBC tumors, we observed significant enrichment of gene signatures associated with the normal developmental processes of pregnancy and weaning-induced breast involution, which supports the idea that PPBC tumors are durably influenced by their host environment (Fig. 2e, f). To further investigate the potential role of normal postpartum biology in the imprinting of tumor biology, we next explored the relationship between our PPBC cases and gene expression signatures obtained from whole-transcriptome profiling from breast tissue of healthy patients ($n = 109$)[56]. We analyzed this publicly available data set to focus on gene sets from healthy nulliparous and postpartum subjects

**a** **Cell Cycle-Proliferation**

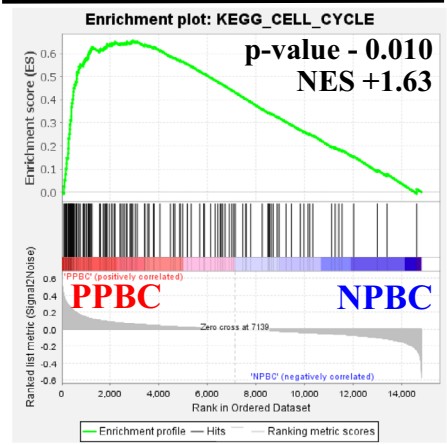

**b** **Cell Death-DNA Repair**

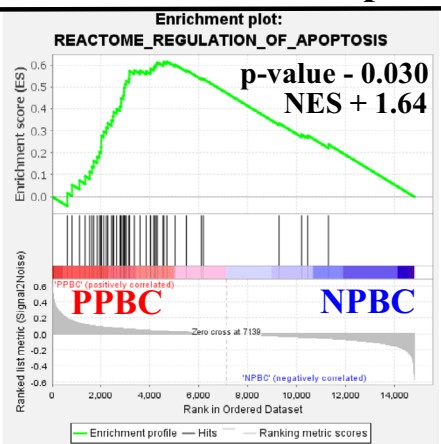

**c** **T-cell Presence - Activation**

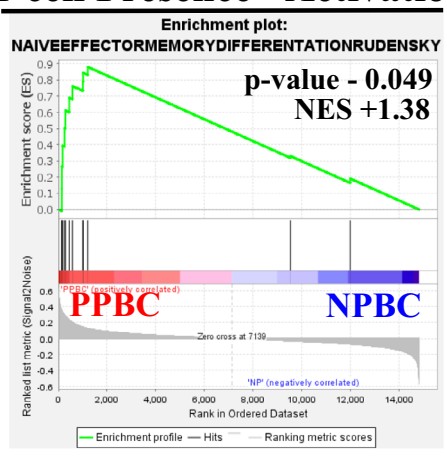

**d** **ER (-)  Breast Cancer**

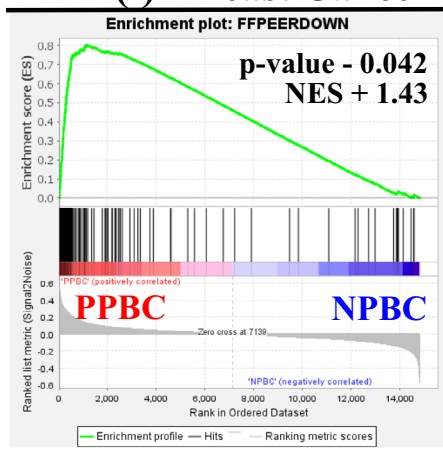

**e** **Mammary Gland Involution**

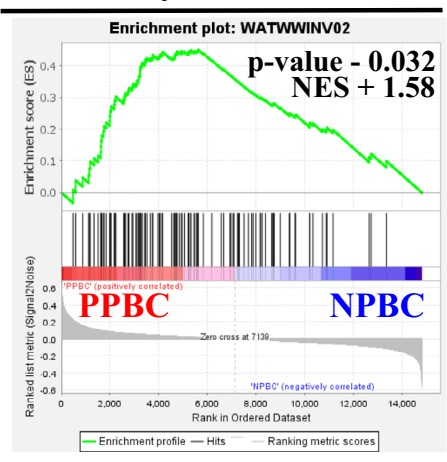

**f** **Parity**

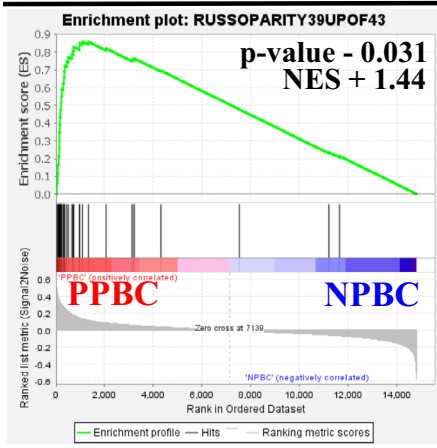

within 2 years of their last childbirth. As anticipated from previous reports[56,57], we observed some normal involution signatures in the postpartum normal tissue expression data sets, such as a parity signature (Supplementary Fig. 1a) and the immune infiltrate signature (Supplementary Fig. 1b). However, neither the immune exhaustion signature (Supplementary Fig. 1c), the ER-negative breast cancer signature (Supplementary Fig. 1d) nor the proliferation signatures (Supplementary Fig. 1e, f) were upregulated in normal postpartum tissue, whereas these gene signatures were upregulated in PPBC samples. One interpretation of these data is that PPBC is a convergence between breast cancer and the reproductive milieu.

**Fig. 2 GSEA identifies cell cycle, cell death, T-cell immunity, estrogen receptor signaling, and mammary gland developmental gene sets as differentially expressed between PPBC and NPBC.** Gene Set Enrichment Analysis (GSEA) was performed on normalized RNA Seq expression data from biologically independent samples of postpartum breast cancer (PPBC, red, n = 9) and nulliparous breast cancer (NPBC, blue, n = 7) patients from Fig. 1, utilizing Molecular Signature Database Collections (V. 7.0) and 100 custom gene lists compiled from the literature review. Gene sets with p values < 0.05 belonging to six biological processes were manually curated: **a** cell cycle and proliferation, **b** cell death and DNA damage repair, **c** T-cell related immunity, **d** estrogen receptor signaling and estrogen receptor-negative breast cancer, **e** post-lactation mammary gland involution in rodents, and **f** parity status in the human breast. Representative enrichment plots from each group are displayed with the determined nominal (non-adjusted) p value and normalized enrichment score (NES)[103].

**Proliferation and TP53 characteristics of PPBC.** To further explore the relationship between the observed cell cycle gene signature upregulated in PPBC and tumor cell proliferation, we examined additional cell cycle gene sets and performed immunohistochemistry staining for the cell cycle protein KI67. While multiple gene sets (Fig. 3a) and single sample composite gene score analyses (Fig. 3b) confirmed statistically significant enrichment of cell cycle genes in PPBC, IHC staining for KI67 did not differ by parity status in our FFPE RNA Seq samples (Fig. 3c, circles, *pseq* = 0.3754). To more rigorously assess the consistency of our RNA Seq findings, we expanded our IHC cohort to include additional young women's, ER+ PPBC and NPBC, FFPE specimens (Fig. 3c, squares) providing 15 samples in each group. With this expanded cohort we found no statistical significance in KI67 staining between these groups (p = 0.3325), depicting a disparity between protein single stain of proliferation (KI67) and composite gene evaluations of proliferative activity. We next explored the signature of increased cell death, DNA damage, and DNA repair gene signatures in PPBC (Fig. 2c), which could suggest increased genetic instability in PPBC tumors. Additional pathway analyses found elevated programmed cell death and TP53 pathways in PPBC (Fig. 3d), data consistent with mutant TP53. To address this possibility, we utilized expression profiling sequences to perform genomic analysis toolkit (GATK) mutational calling, followed by cross-referencing for known TP53 mutations[58] (Fig. 3e, flow chart). These analyses identified four out of nine PPBC samples as containing canonical TP53 mutations (Fig. 3e, bar chart). IHC analyses for P53 on all 30 IHC samples validated these mutation calls. Specifically, the four samples with TP53 mutations displayed enhanced P53 staining consistent with stabilization of P53 protein by mutation (Fig. 3e, inset). Of note, within our entire cohort, we observed significant staining (>10% + nuclei) for P53 in most cases. However, staining was not statistically different between PPBC and NPBC samples. To assess the degree these TP53 mutations were responsible for the increased proliferation signature attributes observed in PPBC, we tracked the position of these bona fide TP53 mutants throughout our analysis (orange-filled circles), and found that TP53 mutational status does not correlate with cell cycle score (Fig. 3b), nor KI67 (Fig. 3c).

**PPBC is enriched for T-cell immunity.** The most dominant gene signature identified in PPBC is immunity (Fig. 1c), specifically T-cell presence and activation (Fig. 2c). An important direct mechanism of anti-tumor immunity is direct tumor cell lysis by cytotoxic cells. Thus, we evaluated for cytotoxic cells in the individual cases using a validated gene signature[59,60], referred to as an "immune infiltrate" signature, which is reflective of the presence of cytotoxic T-cells or NK cells. We observed PPBC samples were enriched in this immune infiltrate signature (Fig. 4a). We next considered that PPBC tumors might be overall enriched for immune cells; however, examination of CD45 (a pan immune cell gene/protein) by IHC analyses (Supplementary Fig. 2b, p = 0.108) or RNA expression (Supplementary Fig. 2c, p = 0.351) found that CD45 was not significantly increased in

PPBC tumors. These data are consistent with specific enrichment of cytotoxic immune cells or T-cells within PPBC. To further delineate between these possibilities we performed T-cell receptor (TCR) repertoire analysis to look for T-cell number and evidence of activation. Using RNA Seq expression data, TCR repertoire analysis revealed more unique TCR sequences from PPBC compared with NPBC samples (Fig. 4b). Increased TCR repertoire could be the consequence of increased diversity of tumor resident T-cell clones or the consequence of having increased overall T-cell numbers in PPBC specimens. To address the relative diversity of the repertoire, we performed normalized clonal analyses. The normalized entropy (clonality index) analysis (Fig. 4c) and the Gini index analysis (Supplementary Fig. 2d) are different mathematical models which both assess the diversity of the repertoire relative to overall numbers of unique TCR sequences. Both of these normalized measures of TCR diversity depict a reduced TCR diversity in PPBC specimens, indicative of clonal expansion. Further, we observe the increased clonality to occur in PPBC within the "hyper-expanded" and "small" frequency population of T-cell clones (Supplementary Fig. 2e). Collectively, increased TCR sequences with increased clonality in two different clonal space populations implicate T-cell activation, which could occur through expansion of tissue-resident memory populations as a consequence of inflammation and/or by antigen-specific T-cell responses[61,62]. Overall, these data are consistent with PPBC tumors eliciting a stronger T-cell response (immunologically hotter) when compared with NPBC.

Greater insight as to how a T-cell presence may influence the tumor microenvironment and perhaps contribute to the response to therapy can be gained by a better understanding of attributes of the T-cell pool. Interestingly, in our GSEA analysis, one of the significant signatures to distinguish between PPBC and NPBC samples was derived from molecular distinctions between exhausted and non-exhausted T-cells found to be conserved between chronic viral and tumor murine models (Fig. 4d). Given the potential importance of this exhausted T-cell enrichment profile, we performed additional analyses to further understand the nature of the T-cells in PPBC samples. First, we performed CIBERSORT[63] analyses (Supplementary Data 2), which provides a normalized estimation of specific immune cell populations from mixed population RNA expression data. CIBERSORT analyses reported significantly (p = 0.009) increased levels of CD8 T-cells in PPBC cases (Fig. 4e), which we confirmed by IHC analyses (Supplementary Fig. 2f). CIBERSORT also reported a significant increase in T follicular helper cells (Tfh) (Fig. 4f). Interestingly, among the molecules that distinguish Tfh from other T-cell populations is the high expression of PD-1[64]. Although widely utilized, CIBERSORT has demonstrated limitations in accurately predicting differential abundance amongst cell populations with similar features. To more robustly characterize the abundance and identity of T-cells in PPBC compared with NPBC we performed mIHC staining with a specific emphasis on PD-1 (a shared feature of activated, exhausted, and Tfh T-cells) and the exhaustion correlated transcription factor TOX1[65–67]. One distinct advantage to mIHC and image cytometry is the ability

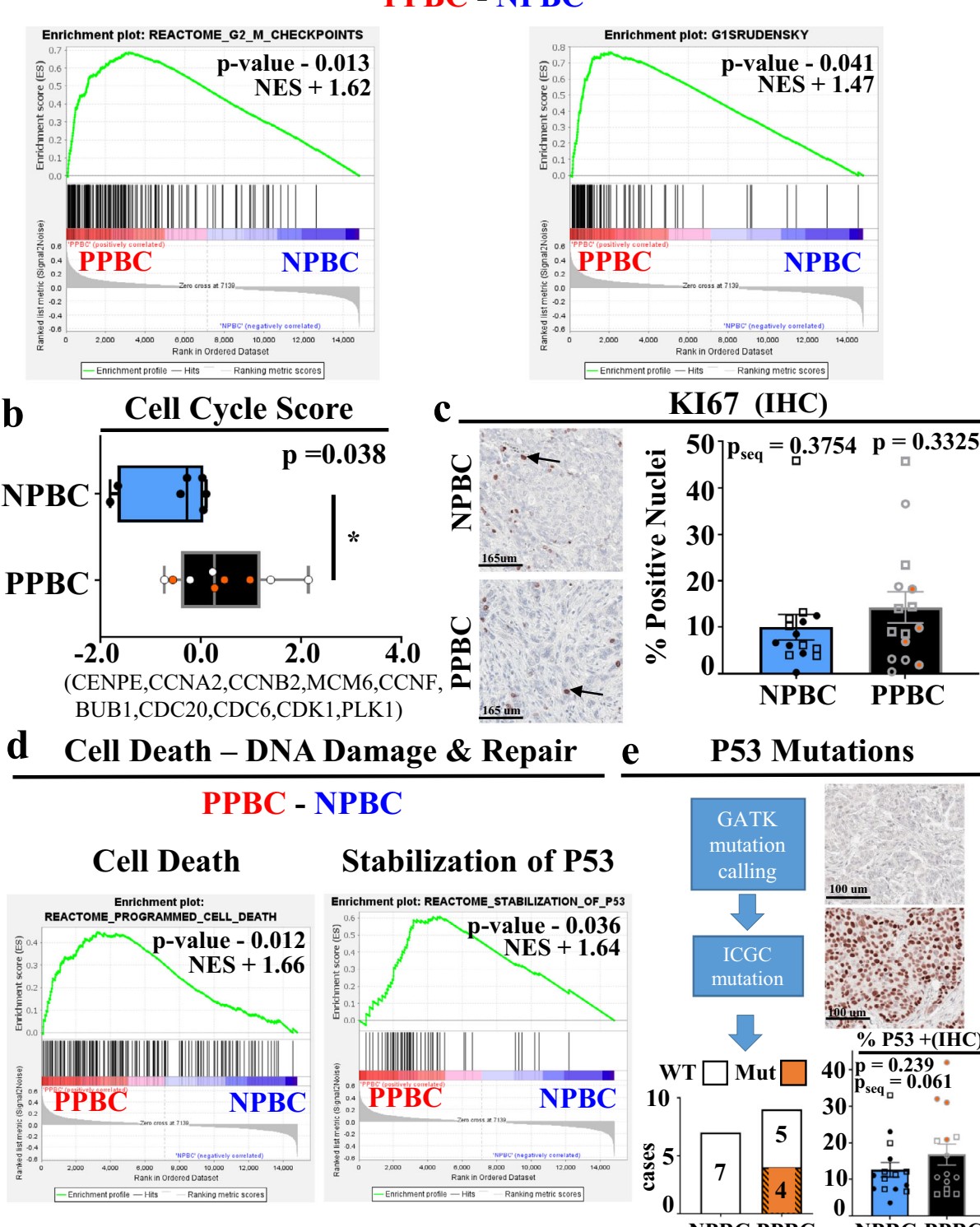

**a** Cell Cycle - Proliferation

**b** Cell Cycle Score

**c** KI67 (IHC)

**d** Cell Death – DNA Damage & Repair

**e** P53 Mutations

to deepen subset analyses based upon the context of intact tissue. In our samples, we noted a prominent accumulation of T-cells (CD3+) at the tumor border (Fig. 4gi, Source Data) in both PPBC and NPBC. When this tumor border region was interrogated by image cytometry, we identified a statistically significant increase of T-cells—and more specifically of CD4 T-cells—as a fraction of all immune cells (CD45+, Fig. 4gii-v, CD3 $p = 0.046$, CD4 $p = 0.014$). Regarding the relative polarization and activation of T-cells as evaluated by the expression of PD-1 and TOX1, we observed approximately twofold increases in PD-1+ (red or

**Fig. 3 Cell Cycle and TP53 gene signatures, TP53 mutational analysis, and immunohistochemical validation.** Detailed examination of the proliferation, cell death, and DNA damage pathways, as identified by RNA expression profiling described in Fig. 2, and IHC examination of these pathways. Depiction of **a** two additional GSEA enrichment plots for cell cycle. **b** Single sample cell cycle score determined from RNA expression values from the indicated genes (PPBC $n = 9$, NPBC $n = 7$). Data are presented as a minimum to maximum with median value marked by a line within the depicted interquartile range, and $p$ value determined by Students' unpaired two-tailed $t$ test with Welch correction. **c** Examples of immunohistochemical (IHC) evaluation of KI67-positive (brown color) protein expression (left), with quantification of KI67 signal evaluated as the proportion of nuclei (right, PPBC $n = 15$, NPBC $n = 15$) Data are presented as mean values ±SEM, and $p$ value determined by Students' unpaired two-tailed $t$ test with Welch correction. Samples evaluated by RNA Seq are depicted by circles and pseq refers to p values for these samples only, while expanded cases for IHC are depicted by squares and p values reflect values for the whole cohort. **d** GSEA analysis assessments of cell death (left) or DNA damage and repair associated gene sets (TP53, right) (PPBC $n = 9$, NPBC $n = 7$). **e** Flow diagram outlining computational steps and results for prediction of the presence of wildtype (WT) or mutant (MUT) TP53 genes in PPBC ($n = 9$) and NPBC ($n = 7$) cohorts utilizing RNA Seq expression data (left), and P53 protein expression (brown color) assessed by IHC (PPBC = 15, NPBC = 15), with P53 signal reported as percent positive area (right). Data are presented as mean values ±SEM and $p$ value assessed by students' unpaired two-tailed $t$ test with Welch correction. International Cancer Genome Consortium (ICGC) identified TP53 mutations are noted by orange-filled circles. For GSEA plots, $p$ values and normalized enrichment score (NES) were determined by GSEA software[103] comparing PPBC (red, $n = 9$) and NPBC (blue, $n = 7$) biologically independent samples as described in Fig. 1 and 2.

yellow bars, black star) and PD-1+ TOX1+ T-cells (yellow bar, yellow star) within both the CD4 (Fig. 4giii-iv, white arrows, Fig. 4gvi) and CD8 T-cell (Fig. 4giii-iv, black arrows, Fig. 4gviii) compartments. Intratumoral T-cells were also evaluated; however, in general infiltration beyond the tumor border was sparse. Although these data trended towards the same patterns as observed at the tumor border, the scarcity of populations reduced the numerical power necessary for statistical significance. Combined, these data support the conclusions derived from RNA expression signatures, chiefly that PPBC has increased levels of activated T-cells that express PD-1 and TOX1, which likely contribute to the enhanced signatures of exhaustion from GSEA analysis and the Tfh profile observed from CIBERSORT analyses.

**PPBC regulon activity predicts poor outcomes in YWBC.** To further compare differences between PPBC and NPBC, we assessed transcription factor activity networks known as regulons, as prior work relying on FFPE tissues demonstrated enhanced fidelity of RNA pathway analysis through regulon analysis[55]. Consistent with the STRING and GSEA data above, we observed the most upregulated regulons to be transcription factors associated with cell cycle pathways (e.g., E2F1, E2F4) (Fig. 5a). Second, we noted the most downregulated pathways in PPBC to be TP53 and ESR1, data also consistent with our pathway analyses (Fig. 3e, Fig. 2d). Although all tumors in our study are highly ER+ by clinical assessment and do not differ in percent ER positivity between groups (Supplementary Fig. 3a, b), several of the most differentially regulated regulons between PPBC and NPBC are transcription factors that are also differentially regulated between ER− and ER+ breast cancers[55] (Fig. 5a, green boxes). This correlation becomes more evident when we plot regulon activity in PPBC vs NPBC in comparison with regulon activity previously reported between ER− vs ER+ cases (Fig. 5b)[55]. To evaluate ER signaling further, we plotted the single sample regulon activity score for the ER-associated pathway (ESR1) for all samples, which revealed significantly decreased ESR1 signaling in PPBC compared with NPBC (Fig. 5c).

Several gene sets and weighted gene expression algorithms exist for ascribing tumor cell molecular subtype identity and treatment recommendations, which historically have focused on HR activity as a target for therapy and delineator of subtype. We next evaluated whether these validated gene sets could distinguish between PPBC and NPBC cases. First, we performed PAM50® molecular subtype determination on all 16 samples. Unsupervised hierarchical clustering based upon the PAM50® gene expression values did not robustly separate the 16 cases by parity status or molecular subtype (Fig. 5d). However, traditionally good prognostic luminal A cases in the PPBC group clustered with the poorer prognostic luminal B cases in the NPBC group (red boxes), data consistent with the idea that luminal

A PPBC has poorer outcomes than predicted based on their luminal A designation. Next, we used normalized RNA Seq expression values from Oncotype Dx® genes, designed to provide a recurrence score in ER+ tumors[68,69] to compute pseudo-Oncotype Dx® scores[70,71] (Fig. 5e). As predicted, the Oncotype scores were lowest in luminal A (dark purple), increased in luminal B (pink), with further increases in Her2 (green) and finally basal cases (orange, PPBC only). We also observed a statistically significant increase ($p = 0.034$) in overall Oncotype Dx® score in the PPBC cohort compared with the NPBC cohort, data consistent with overall reduced ER signaling in the PPBC tumors. Likewise, we evaluated how genes in the Mammaprint® signature, which is considered to be a tumor cell-intrinsic determination of tumor cell subtype, devoid of stromal-related genes, clustered our PPBC and NPBC cases (Supplementary Fig 2c). We found no association between the expression of these genes and the parity status of samples. Combined, these analyses reveal a need for improved prognostic gene signatures for YWBC. We next utilized our results characterizing PPBC through regulon analysis and immune exhaustion gene sets to establish a PPBC signature for ER+ disease.

To generate a composite PPBC signature, we added together the single sample regulon values for the immune exhaustion and E2F1 regulons (the two most upregulated PPBC regulons), and then subtracted the P53 and ESR1 regulon values (the two most downregulated PPBC regulons). To determine whether this PPBC gene expression signature could predict outcomes in an ER+ YWBC cohort. We assembled a YWBC cohort (≤45 years old) with outcomes data by compiling gene expression data across seven previously published studies[72–78]. Although no parity history was available on these publicly available cases, upon applying our PPBC gene signature to this YWBC cohort ($n = 311$ patients with both ER+ and ER− disease) we observed a highly significant decrease in 15-year overall survival in breast cancer patients with a PPBC Hi signature score (HR 2.134, $p = 0.0011$) compared with those with a low score (PPBC Lo, Fig. 5f). Classically, ER+ breast cancers have a better prognosis than ER− cancers, and this was found to be true in this cohort as well (Supplementary Fig. 3d, HR = 2.455, $p = 0.0001$). To determine whether our PPBC signature was indicative of only ER status, we repeated the analysis on only the ER+ cases ($n = 214$) and again found statistically significant reduced survival in the PPBC signature high group compared with the low group (Fig. 5g, HR = 2.30, $p = 0.0084$).

## Discussion

In the present study, we addressed whether PPBC is molecularly distinct from breast cancer diagnosed in nulliparous women. We utilized a small FFPE breast cancer cohort, rigorously controlled

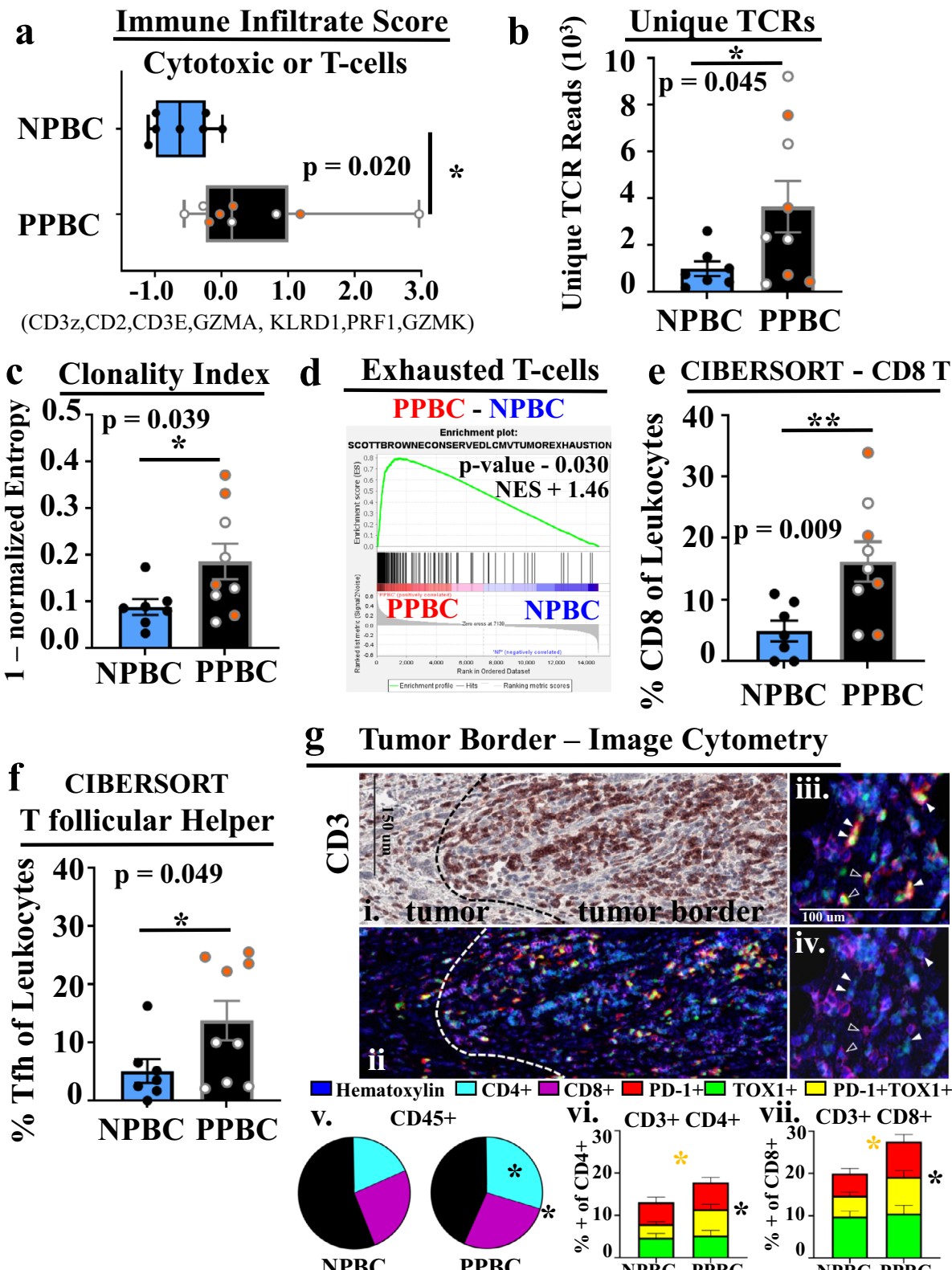

for patient age, BMI, parity history, tumor clinical stage, ER status, and treatment naivety, which permitted us to delineate the role of recent childbirth on tumor gene expression in the absence of potential treatment effects. We observed gene expression signatures of PPBC to include pronounced T-cell presence and

T-cell activation/exhaustion signatures, reduced TP53 activity, reduced ER signaling, and increased cell cycle gene signatures. Further, we find PPBC cases in our cohort are characterized by gene expression signatures associated with normal murine mammary gland involution[79–83], as well as recent childbirth in

**Fig. 4 PPBC is enriched for activated T-cells compared with NPBC.** Characterization of the immune cell presence and T-cell immunity enriched in postpartum breast cancer (PPBC) compared with nulliparous breast cancer (NPBC). **a** RNA Seq expression data was evaluated for genes associated with cytotoxic or T-cell immunity using a gene signature called the single sample immune infiltrate score (PPBC $n = 9$, NPBC $n = 7$). **b** Unique numbers of T-cell receptors (TCR) for each RNA Seq sample, compared between groups (PPBC $n = 9$, NPBC $n = 7$). Relative clonality demonstrated by **c** normalized (norm) entropy (PPBC $n = 9$, NPBC $n = 7$). **d** GSEA profile-derived from exhausted T-cell signature. CIBERSORT deconvolution of RNA Seq data depicts increased **e** CD8 T-cell and **f** T follicular helper (Tfh) presence as a fraction of total leukocytes (PPBC $= 9$, NPBC $= 7$). **g** multiplex IHC analysis of the tumor border in NPBC ($n = 13$) and PPBC ($n = 14$) cases was subjected to quantification by image cytometry. **gi** Hematoxylin (blue) stain & AMEC (red/brown) for CD3 demonstrating T-cell accumulation in the tumor border region. Dashed lines indicate demarcation of intratumoral and tumor border regions. **gii** Aligned pseudo colored multiplex IHC images depicting staining from hematoxylin (dark blue) and chromagen mediated antibody detection of CD4+ (light blue) CD8+ (purple), PD-1+ (red) or TOX1+ (green) cells. PD-1+/TOX1+ cells **giii** appear yellow due to overlap of red and green coloring. **giv** PD-1+ and TOX1+ cells depicted in giii can be either CD4+ (white arrows) or CD8+ (black arrows) T-cells. **gv** Pie-charts depicting increased CD4+ CD3+ T-cells (light blue, $p = 0.0052$) and total T-cell content (light blue and purple, $p = 0.0225$) as fraction of CD45+ cells in the PPBC cohort. PD-1+ (red), TOX1+ (green) or PD-1+ TOX1+ (yellow) cells as a fraction of the **gvi** CD45+ CD3+ CD4+ (yellow group comparison = yellow star $p = 0.0225$, PD-1/red + yellow comparison = black star $p = 0.05$) or **gvii** CD45+ CD3+ CD8+ (yellow group comparison = yellow star $p = 0.0205$, PD-1/red + yellow comparison = black star $p = 0.028$) T-cell compartments. $P$ values determined by GSEA software (nominal, non-adjusted $p$ value, **d**) or Students' unpaired two-tailed $t$ test with Welch correction (**a–c**, **e**, **f**), or Students' unpaired one-tailed $t$ test with Welch correction for confirmatory IHC(*$p \leq 0.05$, **g**). ICGC identified TP53 mutations are noted by orange-filled circles. For box and whisker plot (**a**) data are presented as minimum to maximum with median value marked by a line within the depicted interquartile range, whereas data depicted in bar graphs (**b**, **c**, **e**, **f**, **gvi**, **gvii**) are presented as mean values ±SEM.

healthy women[56,84]. We compiled a signature composed of transcription factor regulons representing the discrete biological pathways differentially expressed in ER+ PPBC and applied this PPBC regulon signature in a large YWBC population. This analysis revealed a significant overall survival disadvantage in young women who had a high PPBC score compared with those with a low score. In sum, these data are consistent with the transient event of normal mammary gland involution durably influencing breast cancer biology, leading to more lethal cancers.

Our data related to a pronounced T-cell presence and activated/exhausted T-cell signatures in PPBC samples is consistent with the idea that normal weaning-induced breast involution impacts the tumor immune milieu. Normal mammary gland involution is characterized by increased T-cell infiltrate[56], which in rodent models includes regulatory (Foxp3, Il-10) and anergized/tolerized T-cell phenotypes[40,45,50]. Physiologically regulated T-cell suppression likely mitigates the potential for self-antigen recognition that could result during the physiologically normal, massive epithelial cell death phase that occurs with cessation of weaning[85,86]. In rodents, PPBC tumors, but not tumors arising in nulliparous hosts, were characterized by an immune milieu consistent with T-cell suppression and tumor cell immune avoidance[50]. This result is consistent with involution durably altering the tumor immune milieu.

Our observation of loss of wildtype TP53—specifically in PPBC tumors—may also reflect normal, weaning-induced involution biology. The P53 tumor suppressor has been studied extensively with respect to its role in maintaining genomic stability[87]. However, P53 is also established as a physiological regulator of involution where its activation initiates apoptosis in the secretory epithelium[88,89]. We speculate that tumor cells present in the involution environment may obtain a survival advantage by suppressing response to this physiologic TP53 dependent cell death pathway. Of potential relevance, studies comparing early and late age at first pregnancy found that early age at first birth associates with long-term protection, whereas late age at first birth is associated with increased risk for breast cancer. In these studies, TP53 mutations were enhanced in late parity cancer cases[90], implicating older maternal age as an additional risk factor for harboring TP53 mutations. Collectively these results and our observations in the present study warrant further investigation into the relationships between parity, maternal age at first childbirth and P53, in conferring poor prognosis.

A dominant molecular distinction in our genomic cohort data was reduced ER signaling in PPBC cases as compared with NPBC. This observation was surprising given that immunohistochemical assessments revealed these tumors to be highly ER-positive. One simple interpretation of these data is that in the postpartum setting, ER-positive breast cancer is more analogous to ER-negative disease with respect to downstream ER signaling pathways. Consistent with our observations of reduced estrogen signaling in PPBC, in a study of postpartum normal and tumor breast tissue[84], the signatures of ER signaling (ESR1) were reduced in postpartum cases compared with their nulliparous counterparts[91]. As with P53, it is possible that the downregulation of ER signaling in tumor cells is a specific adaptation to the involution microenvironment. Signal transducer and activator of transcription (Stat) 5a is a well-established positive regulator of lactation and its suppression is a requisite for the execution of epithelial cell death after weaning[80]. Further, Stat5 expression is under estrogen control in the murine mammary gland[92]. Thus, one untested possibility is that ER+ tumor cells maintain Stat5 survival signaling during involution by downregulating ER signaling. Consistent with this hypothesis, expression of a constitutively activated variant of Stat5 in the murine mammary gland prevented weaning-induced involution and was associated with ER+ adenocarinomas[93]. In sum, our study adds to a growing body of literature reporting poor prognosis in breast cancers expressing classic weaning-induced mammary gland involution gene signatures[56,81,83], and for the first time, extends these studies to demonstrate enrichment of these signatures in breast cancers that have experienced the involution microenvironment.

We also observed a robust increase in cell cycle genes in PPBC. It is noteworthy that these proliferation-associated gene expression signatures did not correlate with increased tumor cell proliferation, as measured by KI67. The lack of increased proliferation in PPBC compared with NPBC is concordant with published data from a large retrospective study showing increased metastasis rates in PPBC compared with nulliparous cases, but similar tumor cell proliferation rates[34], which were also assessed by KI67 protein expression. It is possible that the biology captured in the cell cycle gene sets is, in fact, distinct from cell division biology, and/or that KI67 does not adequately capture cell proliferation[94,95]. Additional research is required to address this apparent conundrum.

Finally, we suggest the gene expression signatures outlined here in human PPBC will provide insight as to why PPBC patients have poorer treatment responses and stimulate interest in alternative treatment approaches. When we considered how our

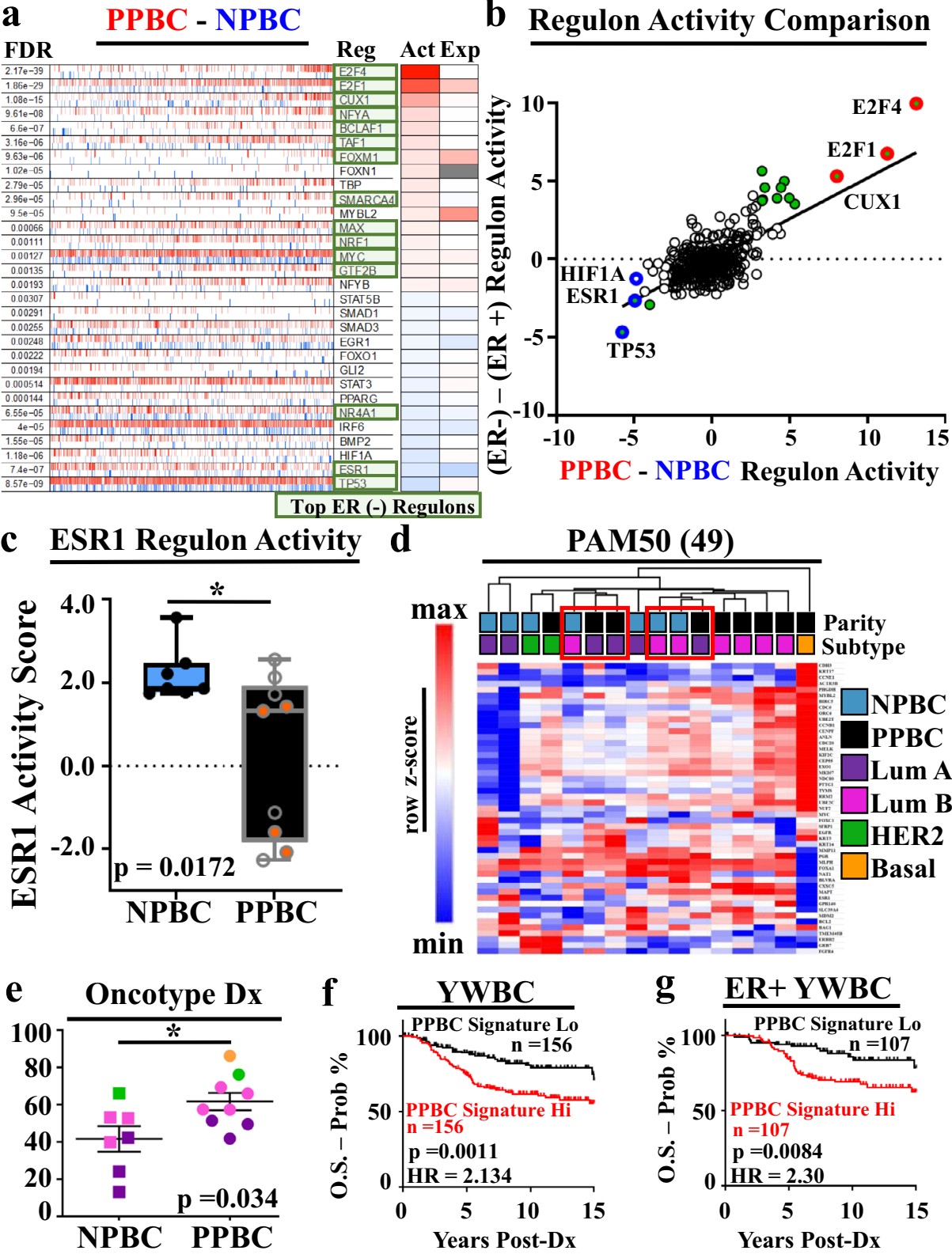

observations fit into existing paradigms of informative gene sets, we found no clear correlation from PAM50® subtype determination nor the Mammaprint® signature. The Oncotype Dx® recurrence score calculation did modestly delineate between nulliparous and PPBC cases, however, the majority of these 16 YWBC cases had high recurrence scores regardless of ER expression or parity status. Thus, further research is needed to determine the best clinical tools capable of delineating low- and high-risk ER+ YWBC and the influence of parity status on those outcomes. By combining parity, treatment, and outcomes data

**Fig. 5 Regulon activity signatures identify key biological processes in PPBC that predict poor prognosis in Young Women's Breast Cancer.** RNA expression profiles between postpartum breast cancer (PPBC, red, $n = 9$) and nulliparous breast cancer (NPBC, blue, $n = 7$) were evaluated for **a** transcriptional network activity through regulon analysis and **b** compared with regulon results comparing FFPE derived ER-negative to ER-positive breast cancer specimens. Most differentially active ER- regulons are highlighted as green boxes (a) and green circles (b). Most differentially active regulons between PPBC and NPBC are highlighted by bolded circles in red (upregulated) or blue (downregulated). **c** Single sample ESR1 regulon activity scores for PPBC ($n = 9$) and NPBC ($n = 7$) were evaluated. ICGC identified TP53 mutations are noted by orange-filled circles. Data are presented as a minimum to maximum with a median value marked by a line within the depicted interquartile range. **d** Gene expression values for the expressed (49) genes of the PAM50® subtype determination assay were evaluated to determine intrinsic subtype for each sample assessed for determination of sample clustering in PPBC and NPBC samples. **e** Pseudo-Oncotype Dx® recurrence scores were derived from RNA expression values for each sample and compared between cohorts (NPBC $n = 7$, PPBC $n = 9$). Data are presented as mean values ±SEM. **f** A postpartum breast cancer regulon-based gene expression signature was composed incorporating immune exhaustion (Fig. 4d), proliferation (E2F1), P53, and ESR1 regulon activity values and evaluated for prognostic significance from a multi-study accumulated cohort of Young Women's Breast Cancer ($n = 311$) composed of female breast cancer patients whose primary breast cancer diagnosed occurred at the age of 45 or under. **g** Subset analysis in ER-positive cases ($n = 214$) from this YWBC cohort. Cohorts were split into PPBC signature high (hi, red, $n = 107$) or low (lo, black, $n = 107$) cohorts based upon the median value of the group plotted. *P* values were determined by Students' unpaired two-tailed *t* test with Welch's correction (**c**, **e**) or by two-tailed log-rank (Mantel–Cox) evaluation for survival plots (**f**, **g**). Log-rank evaluated Hazard Ratios (HR) are depicted.

already available, it may be possible to inform novel treatment strategies for PPBC and determine if any of the existing agents for overcoming ER therapeutic resistance, such as the CDK4/6 inhibitors and their inhibition of the cell cycle, may have added benefit for PPBC, or identify other novel combinations. In addition, given the observation that PPBC, which evolved in the involution environment, has an elevated and activated T-cell compartment with increased expression of PD-1, there may be a select benefit for these patients from checkpoint blockade inhibitors. Already, preclinical data in mouse models depict unique and favorable responses in PPBC tumors to immune modulation via COX-2 suppression[38,96] as well as checkpoint blockade[97].

The chief limitations in this study are the modest size of the NPBC and PPBC cohorts and the reliance on FFPE tissues. As recently highlighted, both of these limitations are predicated on the lack of well-annotated clinical data in YWBC, including time since last pregnancy, as well as the relative rarity of YWBC and PPBC[98]. A further limitation is that the immune milieu profiling by mIHC was focused on a small subset of T-cell activation and exhaustion markers. Future studies are needed to better understand the complexity of the immune milieu in YWBC in general, and in PPBC specifically. Studies of PPBC utilizing fresh, and therefore potentially more informative specimens, necessitate multi-institutional coordination, a worthy objective given the poor prognosis of this disease.

This study utilized an extensive chart review of a single breast cancer repository, spanning 15 years of samples, to build a rigorously controlled FFPE cohort of YWBC with known reproductive histories. This approach demonstrated that ER+ breast cancer in the background of recent childbirth is a molecularly distinct, poor prognostic subtype. This study serves as a molecular anchor point, aligned with extensive epidemiological data, which can support future studies focused on the utilization of fresh samples and larger cohorts. Such studies will undoubtedly provide further insights into the interactions between reproductive history, breast cancer biology, and YWBC patient outcomes, with the potential to improve clinical practice and patient outcomes.

## Methods

**Ethics approval and consent.** The research was conducted on archived FFPE tissues samples collected under IRB-approved protocols at the Kaiser Permanente Northwest Center for Health Research (KPNW IRB) and the Oregon Health & Science University (OHSU IRB). These tissue archives are comprised of clinical samples obtained from women with invasive cancer who were receiving standard of care treatment. The study was retrospective, entailing the use of routinely collected data and archival invasive breast disease tissue and therefore granted a waiver of informed consent by the participating IRBs. All data were fully anonymized before

access by the researchers, labeled only with study-specific identifiers at all points, and the study was approved by the Committee on Clinical Investigations of the OHSU and by the Kaiser Permanente Northwest Biospecimen Review Committee.

**Sample description.** Archival FFPE breast cancer tissues ($n = 40$) were from primary breast cancers of premenopausal women aged 21–45. Inclusion criteria for the cases section were based on age at cancer diagnosis (≤45), parity status, body mass index (BMI), and availability of necessary clinical data and archived tissue specimens. The study was open to all races and ethnicities, however, based on study site demographics, the majority of the study population was white, nonhispanic, women (73%) (Supplementary Table 1a). Exclusion criteria included unknown time intervals from last childbirth, cases who were pregnant at breast cancer diagnosis, archived tissue specimens unavailable for research use, or from women who did not give consent for use of their tissue or clinical data for future research. As our study specifically used breast tissue that was naive for any treatment including neoadjuvant therapy, if that tissue was unavailable for research we excluded the case from the current study. Further, ER-negative cases ($n = 9$) and cases with DCIS without evidence of invasive cancer on the available tissue section ($n = 1$) were excluded from the current study. Using the above inclusion and exclusion criteria, the selected cases ($n = 30$) included women under the age of 45, ER-positive, who were either diagnosed with invasive breast cancer ≤4 years of last childbirth (PPBC) or were nulliparous (cases with spontaneous and/or elected abortions were excluded) based on reproductive history recorded in clinical charts (NPBC). The clinical characteristics of this cohort are shown in Supplementary Table 1a.

All archived H&E-stained slides from clinically indicated surgery were evaluated by a pathologist for each case. Blocks from slides with >80% tumor content were chosen for RNA extraction (10 μm sections), and sequential sections were used for immunohistochemical analysis (4 μm sections).

**RNA isolation.** Total RNA was extracted from freshly cut 10 μm FFPE sections using the miRNeasy FFPE kit (Qiagen, Valencia, CA) according to the manufacturer's protocol, using 1–4 sections (10–40 μm) per case[55]. RNA yield was determined by UV absorption on a NanoDrop 1000 spectrophotometer and fragment size was analyzed using the RNA 6000 Nano assay (Agilent Technologies, Santa Clara, CA) run on the 2100 Bioanalyzer. RNA quality was assessed using DV200 values. Of 40 cases meeting our inclusion criteria, 16 ER+ cases (PPBC ($n = 9$), NPBC ($n = 7$)) yielded RNA of quality (DV200 > 27%) needed to advance to RNA sequencing and in-depth RNA expression profiling. The tumor characteristics of these tumors are presented in Table 1b.

**Library preparation and sequencing.** An input of 75 ng of total FFPE derived RNA was used with the TruSeq RNA Access Library Prep Kit and was prepared according to manufacturer instructions (Illumina, San Diego, CA). Libraries were quantified by real-time PCR using KAPA Library Quantification kits (Kapa Biosystems, Wilmington, MA) on ABI StepOne thermocycler, pooled according to library method (three libraries per lane), and sequenced on a Hi-Seq 2500 (Illumina) using a 100 cycle, single-end protocol providing ~90 million reads per sample. Base call files were converted to fastq format using Bcl2Fastq (Illumina), as described[55].

**RNA sequence alignment.** All RNA Seq reads were aligned to the human reference genome (GRCh38, release 84) using STAR (version 2.5.2b)[99] with default parameters. The STAR "GeneCounts" module was used to quantify the number of reads mapping to each gene. We also used RSEM (version: v1.2.31) to quantify fragments per kilobase of transcript per million (FPKM) of the gene expressions.

**Data processing and significance testing**. Gene expressions quantified by read counts from STAR were used as input into DESeq2[100] for differential expression gene (DEG) analysis. Genes with counts per million (cpm) >0.05 in at least three cases for each group were kept (14,830 genes) for subsequent DEG analyses. DEG analysis was performed by comparing the PPBC cases and the nulliparous cases. The differentially expressed genes were called based on the FDR 0.01 and log two-fold change >1. In the DESeq2 package, counts were normalized using the variance stabilizing transformation (VST) module in DESeq2 for downstream analyses.

**Breast cancer subtype prediction**. All cases included in the study were designated as ER-positive as per clinical immunohistochemical evaluation. Using the PAM50® prediction parameters as described by Parker et al.[101], the tumor biologic subtypes (luminal A, luminal B, Basal, HER2) were predicted for these cases based upon gene expression values derived from whole-exome sequencing.

**Mammaprint®, and PAM50® gene set heatmaps**. To assess the ability of previously reported cancer gene sets to distinguished cohorts, VST transformed counts by DESeq2 were a subset for all expressed matching genes from the Mammaprint®, and PAM50® gene sets. Dendrograms were produced using hierarchical clustering of the z score transformed Euclidean average linkage distances through the Morpheus software package (https://software.broadinstitute.org/morpheus). For PAM50®, subtypes, as well as proliferation, ER, and HER2 scores, were generated using the original prediction parameters as described by Parker et al.[101].

**Pseudo-oncotype Dx® score**. Whole-exome sequencing derived rather than clinical diagnostic approved Oncotype Dx® scores (therefore pseudo) were calculated from reported gene expression values utilizing reported normalization equations[70,71,102]. Specifically, all group scores were determined by subtracting the average expression value across control genes (normalized reference) from each target gene value and adding a value of 10 to the difference and scores computed[102].

**Gene set enrichment analysis**. For GSEA[103] on PPBC compared with NPBC, GSEA version 4.0.3 was used to identify enriched gene sets from the Molecular Signature Database (MSigDB v 7.0, Hallmark, Collection 2, 3, 5–7), as well as 100 customized gene sets prepared from studies relevant to breast cancer, cancer immunity and normal breast biology[56,59,80,81,83,84,90,104–113] (Supplementary Data 01). Gene sets were considered to be enriched if their FDR q value was 0.05. Whole-exome gene expression array data from healthy nulliparous (NP, blue, n = 30) or healthy postpartum breast (PP, red, within 2 years of completed pregnancy, n = 10) tissues was obtained from a previous study (GEO Accession#GSE26457) normalized using Transcriptome Analysis Console software (TAC V.4.02, ThermoFisher Scientific). and used as input values for comparison of GSEA profiles (Supplementary Fig. 1).

**Master regulator analysis**. In order to infer the activities of transcription factors, we used the master regulator inference algorithm (MARINa)[114] compiled in R 'viper' package[115] to perform the regulon analyses on PPBC and NPBC samples. Two sources of data, gene expression signature, and regulatory network were required as model inputs. In this work, the Student's t test based statistic as suggested in viper manual was used as gene expression signatures. The regulon used for the transcription factor activity inference was curated from four databases[116]. The single sample-based regulon activities were inferred by function "viper", which is an extension of MARINa[114] and transforms a gene expression matrix to a regulatory protein activity matrix. For the model input, we used the FPKM quantification of PPBC and NPBC samples as the expression matrix and the same regulon network described above as the regulatory network.

**Clonal entropy and reciprocally related clonal index analyses**. We employed MiXCR (Version 3.0.12,MiLaboratory LLC) (Fig. 4d, e) to analyze TCR, which has an option to identify TCRs from standard RNA Seq. Specifically, MiXCR removed out-of-frame TCR sequences and identified unique V-CDR3 (nucleotide sequence)-J seed sequence and clustered identical sequences computing the frequency of each unique TCR clonotype. A number of TCR repertoire metrics were reported by summarizing the results from MiXCR, including the number of unique TCRs, normalized entropy, clonality index, and repertoire occupancy. Diversity was represented by normalized Shannon entropy (H) reflecting a quantitative measure of how many unique TCR clonotypes were present per sample, and simultaneously indicating how evenly they were distributed (p). For diversity measurement, the value of a diversity index increases when the number of unique TCR sequences increases and when evenness increases. For a given number of uniques, the value of a diversity index is maximized when all types of unique TCRs are equally abundant, and calculated using the default entropy function from the entropy R package using the formula: $H = -\sum_{k=1}^{n} f_k \times \ln(f_k)$, where n is the number of unique clonotypes in a sample, k represents a particular clonotype and f is the frequency of the kth clonotype. Clonality or Clonal index (C) reflects the inverse of the normalized Shannon's entropy H, a statistic for how much of the

repertoire is made up of expanded clones calculated by $C = 1 - H/\ln(n)$, where H is the Shannon entropy, and n is the number of unique clonotypes per sample.

**CIBERSORT analysis**. In order to estimate the abundances of immune cells from the bulk RNA Seq, we utilized CIBERSORT[63] to calculate the proportions of 22 human leukocyte cell subsets defined in the CIBERSORT package for each bulk RNA seq sample. Statistical significance of proportions of each immune cell type between NPBC and PPBC were determined using a two-tailed Student's t test with Welch's correction.

**Compilation and analysis of YWBC cohort**. In this study, YWBC data sets were collected from 8 studies and downloaded from the Gene Expression Omnibus (GEO) with the following accession number: GSE1992[72], GSE20624[73], GSE21653[74], GSE6532[75], GSE2990[79], GSE4922[76], GSE7390[77], and GSE19615[78]. The GEOquery and biomaRt R packages[117] were used to download the raw expression and meta data. YWBC was defined as a diagnosis at age less than 45, which resulted in 648 YWBC samples in total. The raw data sets with different Affymetrix platforms were merged together and the expressions of all data sets were corrected by ComBat R package to remove the underlying batch effects. The averaged expression profiles of microarray probe IDs that map to the same gene symbols were used to quantify the gene expressions for these 648 samples.

**Multiplex IHC, Aperio, and ER quantification**. Formalin-fixed, paraffin-embedded (FFPE) tissues were sectioned at 4 μm. Prior to staining, slides were baked for 2 h at 60 °C and then rehydrated through sequential immersion through xylene, graded alcohols, and water. Next, slides were antigen retrieved in a pressure cooker using DAKO Target Retrieval Solution (pH 6) at 125 °C for 5 min and then cyclically probed[50] in the following order with the indicated antibody, dilution and incubation times: Cycle 1 (PD-1, abcam, ab52587, Clone NAT105, 1:100, 1Hr), Cycle 2 (KI67, Thermofisher, RM-9106-S, Clone SP6, 1:300, 1Hr), Cycle 3 (TOX1, abcam, ab237009, Clone NAN448B, 1:800, overnight), Cycle 4 (P53, Thermofisher, MA5-12557, Clone DO-7, 1:100, 2Hr), Cycle 5 (Phospho-Histone H2A.X (Ser139), Cell Signaling, 9718, Clone 20E3, 1:250, 1Hr), Cycle 6 (CD8, BioSB, BSB5174, Clone C8/144B, 1:100, 1Hr), Cycle 7 (CD3, Dako, A0452, 1:400, overnight), Cycle 8 (CD45, Dako, M0701, Clones 2B11+ PD7/26, 1:300, 1Hr). Next, secondary anti-rabbit or anti-mouse Simple Stain MAX PO Histofine Peroxidase Polymer (Nichirei Biochemicals, 414144 or 414134) or anti-rat ImmPRESS Peroxidase Polymer (Vector Laboratories, MP-7444) antibodies were applied, followed by chromogenic detection with peroxidase substrate 3-amino-9-ethylcarbazole (AEC). The stained sections were scanned digitally using Aperio Image Scope AT2 (Leica Biosystems, CA, USA) at 20x magnification. For Aperio analysis, scanned images were visualized on Image scope software (v12.4.3) and the tissue sections were annotated for all tumor areas present per section followed by the semi-quantitative image analysis performed on the entire tumor area using Aperio deconvolution and nuclear algorithms (Leica Biosystems, CA, USA)[57,118]. Further, for the CD45, CD3, CD8, PD-1, and Tox multiplex IHC (mIHC) staining analysis on a per-cell basis, the pixel density of the scanned images necessitates region of interest (ROI) analysis, thus ~3–4 ROIs were selected per case where the immune cell infiltrate was high (based on H&E and CD45 staining review). Regions with high immune cell infiltrate were selected so that sufficient events needed to perform statistically supported single-cell analyses were captured. The selection of ROIs for each case was done by 2 analysts blinded to the reproductive status of the cases, with cases randomly sorted prior to ROI selection. Image processing, alignment of selected regions, and extraction of AEC signals was performed in MATLAB (V9.90.1592791) using the SURF algorithm in the Computer Vision Toolbox (The MathWorks, Inc) and FIJI as reported[55,119]. Pipeline for image processing and cell quantification was performed using FIJI (FIJI v 2.1), CellProfiler Version 4.1.3, and FCS Express Image Cytometry RUO (7.06.0015, De Novo Software, Glendale, CA)[120]. ER staining (ER, Novocastra, NCL-L-ER-6F11, 1:200, 1Hr) and pathological assessment for the intensity and % positive tumor cells for assigning an overall percent positive staining was done by a pathologist. The % positive ER results were independently confirmed by a second observer blinded to the study group. mIHC evaluation was carried out for all cases which passed multiple image alignment and segmentation quality control evaluations (n = 13 NPBC and n = 14 PPBC cases).

**Statistics and reproducibility**. Statistical significance determined by p values were generated by GraphPad-Prism Software (V9.2.0) (GraphPad Software, San Diego, California USA) unless otherwise stated and was performed as Students' unpaired two-tailed t test with Welch correction (*$p \le 0.05$), or Students' unpaired one-tailed t test with Welch correction for apriori directionality in confirmatory IHC. Survival curves were also plotted with GraphPad-Prism Software and p values reported from log-rank (Mantel–Cox) evaluation and log-rank Hazard Ratios (HR) reported.

To preserve the precious human clinical samples the mIHC staining was conducted once. With each staining run, human breast cancer and tonsil tissue samples were used as negative and positive controls for technical validation of staining and standardization of analysis across cases.

RNA library preparation and sequencing were carried out once per case utilizing previously established methodologies which demonstrated reproducibility

of a single technical replicate through evaluation of isolation and sequencing replicates[55].

**Reporting summary**. Further information on research design is available in the Nature Research Reporting Summary linked to this article.

## Data availability

The RNA-derived sequencing data generated in this study have been deposited in the Gene Expression Omnibus (GEO) database under accession code GSE158854. The publicly available RNA expression data from healthy nulliparous and postpartum breast tissues used in this study are available in the GEO database under accession code GSE26457. The publicly available outcomes data based upon RNA expression profiling used in this study as a YWBC cohort are available from the GEO database under accession codes, GSE1992, GSE20624, GSE21653, GSE6532, GSE2990, GSE4922, GSE7390, and GSE19615. All numerical data used in generating plots of figures are available as Source Data. All remaining data are available within the Article, Supplementary Information, or Source Data files. Source data are provided with this paper.

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

## Acknowledgements

This work was funded by grant NIH/NCI R01#1CA169175 to P.S. and V.F.B.; the Willard L. and Ruth P. Eccles Foundation, the Coit Family Foundation, Oregon Health & Science University—School of Medicine Faculty Innovation Funds, and Oregon Clinical and Translational Research Institute (OCTRI)–Kaiser Permanente Northwest tissue retrieval funds, and the Knight Cancer Institute to P.S.; and funding from the Avon Foundation to P.G. We also want to thank Colorado's NIH/NCI Cancer Center Support Grant P30CA046934, the Knight Cancer Institute's Cancer Center Support Grant P30CA69533, NIH-OD011092 for the Oregon National Primate Research Center Bioinformatics and Biostatistics Core, and NIH/NLM K01 K01LM012877 (to Z.X.) and the Collins Medical Trust grant (to Z.X.). The funding bodies listed above played no role in; the design of the study, collection, analysis, or interpretation of data, nor were they involved in the preparation of this manuscript. The authors wish to acknowledge the Gene Profiling Shared Resource and Massive Parallel Sequencing Shared resources at OHSU for oversight for isolation and sequencing of RNA, Kristin Muessig and Chalinya L Ingphakorn (Kaiser Permanente Northwest) for administrative support; Weston Anderson for excellent support in review and editing of the manuscript and Wendy Ingman and Sarah Bernhardt for assistance in computing pseudo-Oncotype Dx® scores. RNA extractions were performed by the OHSU Gene Profiling Shared Resource. Illumina sequencing was performed by the OHSU Massively Parallel Sequencing Shared Resource.

## Author contributions

V.F.B. and P.S. provided the study objectives. N.P., S.J., S.W., P.G., V.F.B., and P.S. designed the study. S.W. and S.J. interrogated the Kaiser Registry for case selection and clinical data set preparation. Z.X. and W.H. performed RNA Seq data alignment, normalization, and quality check. Z.X., N.P., and D.S. performed pathway analyses, and D.S. performed regulon analyses. S.J. and J.N. designed and performed mIHC experiments, J.N and M.O. performed image processing for mIHC. A.B. and M.O. performed image cytometry for single-cell quantification of mIHC data sets and data analysis. N.P., S.J., W.H., A.B., and D.S. generated figures. N.P., S.J., W.H., P.S., and Z.X. interpreted all results, and S. J., N.P., and P.S. composed the manuscript. All authors critically reviewed the manuscript. S.J., N.P., and D.S. contributed equally to the manuscript and without each of their unique contributions, the work could not have been accomplished, and thus share the first author position. P.S. and Z.X. contributed equally to the manuscript and are responsible for data integrity. All authors read and approved the final manuscript.

## Competing interests

The authors declare no competing interests.
