## [Peer Review File · Nature Communications]

Postpartum breast cancer has a distinct molecular profile that predicts poor outcomesReviewers' Comments:

Reviewer #1:

Remarks to the Author:

On the background of a high mortality rate of YWBC patients which is not only explained by ER and HER2 status, and that a postpartum diagnosis of YWBC is associated with a high mortality rate, Jindal et al. set out to explore whether ER+ YWBC in nulliparous (NPBC) or postpartum (PPBC) patients is molecularly distinct. They perform bulk RNA sequencing on breast cancer tissue from a total of 16 matched PPBC and NPBC ER+ young breast cancer patients. Using several bioinformatical methods, the authors compellingly demonstrate that these NPBC and PPBC are molecularly distinct.

Using primarily GSEA, Jindal et al. explore further what the main biological aspects are by which these two cancer types differ. One of the observations they follow up in more detail is that PPBC scores higher on a 'cytotoxic T cell score', which prompted the authors to compare TCR diversity in the two cancer groups and to compare the expression of molecules associated with T cell exhaustion. The depth of the investigation into immune cell composition is however limited. The presented data are - as the authors state - consistent with the hypothesis that PPBC contain higher numbers of tumor-infiltrating CD8 T cells, and that these T cells are exhausted and more clonally expanded. However, alternative hypotheses are not investigated, and formal proof for the hypothesis is not provided.

To obtain a more definitive understanding of the differences in the immune cell compartment between both tumor groups, analyses at the single cell level would be warranted. The authors do provide microscopical analyses, which are an example of possible single-cell analyses, but the potential of these single cell investigations is currently not fully leveraged.

Taken together, the study provides clear evidence that the two tumor types are molecularly distinct. However, the study is purely descriptive and somewhat limited in depth with regards to the characterization of the exact molecular differences, leaving room for alternative hypotheses, as outlined below in the specific points.

Specific points:

1) A higher score in the used 'cytotoxic T cells immune infiltrate score' cannot unambiguously be interpreted as an increased presence of CD8 T cells. The genes that make up this score are a mix of genes expressed by all T cells (i.e. CD8 T cells, CD4 T cells, NKT cells - all of which will also contribute to the TCR sequences), and genes associated with cytotoxicity - which is a property of both CD8 T cells and NK cells. A high score could therefore for example also result from increased NK cell numbers and increased CD4 T regulatory cells.

To gain a better understanding of possible differences in the immune infiltrate, more precise immune cell type signatures or scores would need to be used. Are all T cells enriched, or just CD8 T cells? What about Treg cells? NK cells, NKT cells? Analyses on CD45 expression suggest that the total immune infiltrate is comparable between both tumor types, which would suggest that if some cell types are enriched in PPBC, others might be enriched in NPBC - which are those?

2) The authors detect a higher amount of unique TCR sequences in PPBC. As they mention, this can be a consequence of having a larger T cell infiltrate. This relationship could be explored to some extent by leveraging the observation that there are large differences between patients in the amount of unique TCR sequences detected, and also in the 'cytotoxic T cells immune infiltrate score' values. Is there a correlation between the # of unique TCR sequences a patient has and the value on a refined 'T cell presence score'?

3) The usage of the term 'cytotoxic T cell' is ambiguous. Do the authors refer to all CD8 T cells? Or to only the fraction of CD8 T cells that express cytotoxicity-related genes? Please specify. The issue is

particularly evident when the authors write about 'exhausted CD8+ cytotoxic cell' (Page 5, line 219). Please specify whether you refer here to all CD8 T cells that are exhausted? Or do you refer only to those (few) PD-1+ T cells that also express cytotoxic molecules?

Please also specify how you define 'exhausted' in this context. All PD-1+ cells? Only the PD1+Tox+ ones (which are not actually cytotoxic).

See e.g. Beltra et al. Immunity 2020 (John Wherry lab) and Hudson et al. Immunity 2019 (Rafi Ahmed lab).

4) In the same context, to aid interpretation, please specify which genes are included in the gene sets used for GSEA that are derived from papers and not readily findable in MSigDB, such as the one for 'exhausted CD8 T cells' that has been used. Alternatively, a clear description from which paper and what exact experimental comparison are these derived from.

Given the uncertainty regarding the genes making up the 'exhausted CD8 T cell' signature, it is unclear whether only exhausted CD8, or also exhausted CD4 T cells count towards the score. It is also unclear whether the enrichment for this gene set in PPBC is a consequence of that tumor type having (potentially) a larger CD8 T cell infiltrate, and therefore simply score higher because of CD8 related genes?

More relevant would be the question whether those CD8 T cells that are present in one tumor type are more exhausted than the ones in the other tumor type. This requires T cell analysis at the single-cell level: What fraction of CD8+ cells is PD-1+, what fraction is PD1+Tox+? The authors have some single-cell data from IHC, but those data are merely evaluated by comparing CD8, PD-1 and Tox staining separately, not on a per-cell basis. Furthermore, given that only the merged picture from all fluorescence channels is shown, it is not possible to judge whether the cells that stain for PD-1 also co-express CD8. With the currently presented analysis I do not consider the authors conclusion that 'these data support PPBC being characterized by an increased presence of exhausted T cells' warranted.

5) For the IHC analysis, please specify how many fields of view are evaluated per patient (the amount of stained cells in the single presented field of view for NPBC is very low). Given that the staining pattern in the provided PPBC picture is rather clustered, please describe how the fields of view have been selected to avoid selection bias.

6) While Tox is expressed by subsets of exhausted CD8 T cells, Tox expression per se cannot be interpreted as evidence for exhaustion, since also highly functional effector-memory CD8 T cells can express Tox (Sekine et al, Science Immunology 2020). In line with this, the presented IHC pictures show Tox-expressing cells that do not co-express PD-1. This should be discussed.

7) Page 5, line 211 " The increased clonality of the small population caught our attention as this is unambiguously consistent with antigen-specific T cell responses 60,61 ". I do not think this statement is warranted based on neither the presented data, nor the two references. The data in figure 4f show that in PPBC, smaller clones make up a larger fraction of the repertoire than in NPBC. This does not necessarily mean that the smaller clones are also less even in size than in NPBC. Also, even if they were less even in size, this is no evidence for a more prominent (tumor) antigen-driven clonal expansion in PPBC. Clones might as well be further expanded due to inflammatory signals, and many tumor-infiltrating T cells are not actually specific for tumor(associated) antigens, but rather for previously encountered viral infections (Scheper et al, Nature Medicine 2019 - Ton Schumacher lab)

Reviewer #2:

Remarks to the Author:

The authors of this manuscript performed comparative gene expression profiling of 9 cases of ER+

postpartum breast cancer (PPBC) and 7 cases of ER+ breast cancer from nulliparous (NPBC) women with the goal of identifying differences that could explain the worse outcome of PPBC. The data on outcome differences between breast cancer diagnosed in young women with differences in relation to pregnancy is not conclusive, presumably due to the relatively low number of patients in any given cohort. Understanding biological differences due to parity could potentially help the clinical management of these patients.

Numerous prior studies (reviewed in Korakiti et al. Front Onc 2020) have characterized the gene expression and/or genetic profiles of PPBC to identify biomarkers and predictors of poor outcome. P53, cell cycle-related genes (e.g., MKI67, MYC), and immune-related (e.g., PD-L1) have been shown to be differentially expressed between PPBC and non-PPBC. This current manuscript essentially confirms these prior data with novelty limited to the using only ER+ tumors and the TCR clonotype data.

Specific points:

1. The number of cases analyzed is very few, especially considering that these are premenopausal women with differences in age, ethnicity, and tumor stage. The authors should expand their sample size and if possible, use expression data from prior publications to strengthen their conclusions.
2. The authors performed RNAseq on bulk FFPE tissues, thus, the observed differences can be due to differences in cellular composition, especially differences in stromal and immune cells. The authors do not show H&E images and do not mention if they controlled for cell types in the samples that were used for RNA-seq. They should at least run CYBERSORT like test to see if the predicted cell types are relatively comparable in the two groups and also perform multicolor immunofluorescence for the cell types and markers they conclude are significantly different between PPBC and NPBC cohorts in a larger set of samples.
3. ER+ breast cancers tend to be immune cold. Yet the authors show that based on IHC for CD45 ~10% of all cells are leukocytes (Suppl Fig. 2b), but they do not show the actual data. How was this staining quantified and how could they get such a high %? Also did this does not correlate with their prediction of CD45+ cells based on RNAseq data (Fig 4b) that shows no difference. They have to analyze the same cases using both methods and show how well they correlate. Furthermore, all IHC and IF should be quantitated based on whole slide scans.
4. The TCR clonotype repertoire data is very confusingly described and presented. How can PPBC have ~4 times more unique TCRs (Fig 4c), but still have the same clonotype diversity (Fig 4d) and at the same time have ~2x higher clonality index (Fig 4e)? the authors have to describe and present these data in more clearly.
5. The differences observed in TCR clonotype diversity could be due to differences in overall peripheral blood TCR diversity. The authors should analyze peripheral blood to determine this.

Minor comment:

1. The manuscript text is very dense (both space and text) and not easy to read. It would be great to improve this.
2. It would be helpful if the authors indicate the figure number on the figures.
3. "Unique numbers of T-cell receptor repertoire (TCR)" in legend of Figure 4 makes no sense. The authors presumably meant Unique numbers of TCRs (TCR stands for T cell receptor)?

Reviewer #3:

Remarks to the Author:

Jindal, Pennock and colleagues present a study characterizing the features of breast cancer arising in young women with a focus on the role of parity in disease pathogenesis (NCOMMS-20-41597). The area of study is very important, the manuscript is well written and the claims are very interesting however one major challenge with this study is the small number of cases that were accessible for analysis (9 PPBC, 7 NPBC) and the molecular heterogeneity of the 9 PPBC of which 4 had TP53 mutation. Despite the authors' significant effort to match samples between the cohorts, it is difficult to assess how strong the conclusions are about breast cancers that arise in this particular clinical setting and whether follow on steps such as generating a so called 'PPBC signature' are appropriate in the context of a small and heterogeneous cohort.

It is unclear if some of the observed findings are driven by the preponderance of P53 mutant cases in the PPBC samples that were available. It would ultimately be helpful to compare in the gene set enrichment analyses how P53 wild type PPBC compare with P53-mutant PPBC and with NPBC? It would help in the analyses and plots to indicate which samples are p53 mutant (colouring of dots in the dot plots is a start – for instance in Fig 3b, c and 3e). Were the authors able to assess if any of these patients had a germline mutation of p53 (Li Fraumeni); which samples were P53 mutant and what were the mutations (should be listed in supplementary table 1). Are P53 mutations enriched in PPBC (please see point further below on the use of other samples that were not included in the RNA analysis).

One set of questions relates to how PPBC and NPBC compare with ER+ breast cancers in the general population and whether some of the gene set categories (such as mammary gland involution) that are enriched in PPBC (vs NPBC) would be enriched in ER+ breast cancer in the general (and older) population vs NPBC. It would help the reader if the basis for how this signature was generated were explained in the text and not solely referenced.

The reliance on the scoring of ER IHC from clinical pathology reports seems suboptimal in this study because the information plays an important role in the authors interpretation of their findings. The quantification of the frequency of ER positive tumor cells and perhaps even the level of ER protein expression on a single cell level may contain meaningful information and that information is accessible to the authors using their multiplexed IHC platform. In fact, some of the clinical cases were only scored as positive or negative without providing a percentage - were those 5-10% positive or 100% positive. What is the threshold at the authors' institution? A more careful characterization and assessment of the expression of ER seems therefore advisable.

The authors screened 40 cases but only 16 were deemed useful for RNA studies. However, very likely some of the cases would have been suitable for tissue imaging (P53 IHC to further assess the frequency of P53 mutation in the PPBC vs NPBC groups; ER IHC to see if the levels are the same between the two groups). Having added such cases would have allowed the authors to expand the size of the cohort for some analyses and could have in part helped strengthen some of the conclusions. While the 8-plex tissue imaging as used in this study does provide an orthogonal method for assessing a few of the observations made using gene set analyses of RNA data, there does seem to be a missed opportunity to characterize more carefully these uncommon samples for several reasons.

1. A tumor marker (e.g., keratin) is not included in the panel and therefore the ability to interpret 'tumor cell proliferation' is fraught because the quantification of Ki67 likely includes immune cells and stromal cells which can often be found to represent a fairly large fraction of proliferative cells in samples. Elevated p53 could in principle be used as a tumor marker in the four TP53 mutant cases but the remaining cases have no tumor marker. It is important to note that Ki67 is known to have graded expression (PMID 30067968) so setting gates for determining Ki67+ and Ki67- cells is not straightforward (please see next comment).
2. The approach for analysis is not well described. Restricting analyses to regions of interest seems unwarranted and in addition it does not seem that important information about the analysis was provided in the manuscript – based on what criteria were the regions selected (randomly? informed by

some feature?, restricted to the tumor compartment or the tumor stromal interface?), how many regions were quantified per case, how many cells were analysed, how do the authors know that these are representative fields and that their results would not be different if other ROIs were selected? If whole slide imaging was performed, whole slide analysis seems to be the preferred approach and would allow for a more sophisticated characterization of these samples. How did the authors go about calling positive and negative cells particularly in cases like Ki67 where expression was likely not bimodal but rather continuous/graded? The use of ROIs for analysis seems important in light of data like the trend toward a significant increase in the CD45 population in the PPBC but the difference is deemed non-significant whereas the levels of CD8 are significant. Is this a regional or global effect?

3. A richer set of markers (keratin and ER mentioned above for instance) would have been very useful in this study. The authors have reported the role of COX2 in YWBC but that was not explored. The levels of Tregs was not monitored. Exhaustion is a complex phenotype to characterize on the level of marker expression so the use of additional exhaustion markers such as LAG3 and TIM3 would have been very informative as would activation markers such as granzyme B and perforin (and Ki67). Those markers would allow a more careful evaluation of what appears to be conflicting results from the gene set enrichment analysis – T cell presence and activation in Figure 2c and exhaustion of CD8 T cells in Figure 4g. PD1 on its own is not a marker of exhaustion and can be present during T cell activation (along with Ki67) – did the authors score double positives for PD1 and TOX? Also, the spatial arrangement of the activated and dysfunctional immune cells is not presented - where are these populations relative to the tumor cells ... are they in the tumor compartment, at the interface or at a distance from the tumor border. Are the levels of dysfunctional T cells influenced by p53 status of the PPBC? In an ideal setting the authors would provide their IHC data to the public for analysis, particularly because these are such uncommon samples and therefore of significant public interest.

4. The authors can be more careful with their descriptions – for instance they describe in the discussion a “pronounced T cell presence” ... but the analysis is only for CD8 T cells and does not include CD4 cells for instance.

Additional points/questions:

Presumably “whole genome profiling” of the breast tissue from healthy patients is RNA-seq?

Why did the PPBC cohort not receive hormone therapy? Is that unexpected?

Was gamma-H2AX assessed from the tissue imaging?

ManuscriptID: NCOMMS-20-41597A-Z

Title: In young women's breast cancer, parity status associates with expression signatures of T cell activation, cell cycle, and reduced ER signaling

Response To Reviewers' comments:

Reviewer #1, expert in TCR repertoire analysis (Remarks to the Author):

On the background of a high mortality rate of YWBC patients which is not only explained by ER and HER2 status, and that a postpartum diagnosis of YWBC is associated with a high mortality rate, Jindal et al. set out to explore whether ER+ YWBC in nulliparous (NPBC) or postpartum (PPBC) patients is molecularly distinct. They perform bulk RNA sequencing on breast cancer tissue from a total of 16 matched PPBC and NPBC ER+ young breast cancer patients. Using several bioinformatical methods, the authors compellingly demonstrate that these NPBC and PPBC are molecularly distinct.

Using primarily GSEA, Jindal et al. explore further what the main biological aspects are by which these two cancer types differ. One of the observations they follow up in more detail is that PPBC scores higher on a 'cytotoxic T cell score', which prompted the authors to compare TCR diversity in the two cancer groups and to compare the expression of molecules associated with T cell exhaustion. The depth of the investigation into immune cell composition is however limited. The presented data are - as the authors state - consistent with the hypothesis that PPBC contain higher numbers of tumor-infiltrating CD8 T cells, and that these T cells are exhausted and more clonally expanded. However, alternative hypotheses are not investigated, and formal proof for the hypothesis is not provided.

To obtain a more definitive understanding of the differences in the immune cell compartment between both tumor groups, analyses at the single cell level would be warranted. The authors do provide microscopical analyses, which are an example of possible single-cell analyses, but the potential of these single cell investigations is currently not fully leveraged.

Taken together, the study provides clear evidence that the two tumor types are molecularly distinct. However, the study is purely descriptive and somewhat limited in depth with regards to the characterization of the exact molecular differences, leaving room for alternative hypotheses, as outlined below in the specific points.

Reviewer #1 Specific point 1:

A higher score in the used 'cytotoxic T cells immune infiltrate score' cannot unambiguously be interpreted as an increased presence of CD8 T cells. The genes that make up this score are a mix of genes expressed by all T cells (i.e. CD8 T cells, CD4 T cells, NKT cells - all of which will also contribute to the TCR sequences), and genes associated with cytotoxicity - which is a property of both CD8 T cells and NK cells. A high score could therefore for example also result from increased NK cell numbers and increased CD4 T regulatory cells.

To gain a better understanding of possible differences in the immune infiltrate, more precise

immune cell type signatures or scores would need to be used. Are all T cells enriched, or just CD8 T cells? What about Treg cells? NK cells, NKT cells? Analyses on CD45 expression suggest that the total immune infiltrate is comparable between both tumor types, which would suggest that if some cell types are enriched in PPBC, others might be enriched in NPBC - which are those?

Response: The authors agree with the reviewer that the genes composing the immune infiltrate score in figure 4a contain attributes descriptive of all cytotoxic T cells, and could indeed be either CD4+ or CD8+ cytotoxic T cells, and further this gene list has overlap with other cytotoxic cell subtypes including Natural Killer (NK) and NK T cells. The goal of the authors in composing the figures in the provided manner was to (1) to utilize existing and accepted tools/gene lists established by authorities in the field to properly frame our observations relative to historical and contemporary approaches of the field; and (2) to build towards greater granularity of understanding towards the bottom of the figure. With regards to figure 4a specifically, the authors are utilizing a list of genes established by authorities in the field (Davoli, Uno, Wooten, & Elledge, 2017). This gene list has been helpful in describing anti-tumor immunity in correlating responses to checkpoint blockade immunotherapy. We agree that there are inherent limitations of our data set and in response to the reviewers suggestions, we have revised the results and discussion sections to reflect the potential roles for CD4, NK and NKT cells. To better reflect this limitation, we have also changed the title of the manuscript to, “In young women’s breast cancer, parity status associates with expression signatures of T cell activation, cell cycle, and reduced ER signaling. We think by adding the term “signature” and eliminating the term exhaustion we remove the premise that we are formally demonstrating specific CD8+ cytotoxic T cell exhaustion, which would require functional assays. This point is covered in further detail below (Reviewer 1 specific points 3 &4).

Further, we agree with the reviewer’s request for enhanced clarity as to whether a single or multiple immune cell population drives the PPBC gene signature. As reviewer 1 noted, in Fig 4b of our original submission, we explored this “immune infiltrate” signature by first establishing that overall CD45 (a pan immune cell gene) is not significantly different between the two groups by RNA expression analyses. As requested by the reviewer, we have furthered this inquiry now by performing CIBERSORT analyses. CIBERSORT is an RNA-Seq “virtual-unmixing” algorithm based on gene expression that deconvolutes specific cell populations from bulk RNA-Seq data and then subsequently evaluates the differential abundance of cell subtypes between cohorts. Upon performing this more extensive normalization and evaluating the relative composition within the CD45+ cohort of all samples, utilizing established and broadly accepted immune subpopulation gene signatures (“LM22”), we observed that classically termed cytotoxic, CD8 T cells are the most enriched group within immune cell populations of the PPBC cohort (3.3-fold increase, p-value 0.009 , new figure 4f) followed by M1 macrophages (4.5-fold increase, p-value 0.04, supplemental data 02) and by T follicular helper cells (2.7 fold increase, p-value 0.04865, new figure 4f). The only statistically enhanced group of immune cells in the NPBC cohort through Cibersort analyses was unpolarized (M0) macrophage subset (2-fold enhancement, p value 0.04). We have provided all the data for Cibersort as a supplemental table (Supplemental Data 02). As reviewer 1 notes, Fig 4,c,d,e,and f (original submission) have no way of distinguishing between conventional CD4, conventional CD8 T, or NK T cells, and we have accordingly changed the subtitles of these figures. Further, to fit the progression of the

narrative of increasing specificity of results, we have now inserted our Cibersort results after the clonal space TCR analyses, which we believe leads nicely into the narrative of immune cell protein based IHC that specifically examines CD8 and CD4 T cell frequency and character. Further, we have reviewed and modified the text to more accurately address the specificity or ambiguity of the identities revealed at each step of figure 4, including what other hypotheses could explain the data at each figure. Finally, we have performed the suggested single-cell analyses, via multiplex immunohistochemistry (mIHC), with these results discussed in detail below.

Reviewer #1 Specific Point 2

The authors detect a higher amount of unique TCR sequences in PPBC. As they mention, this can be a consequence of having a larger T cell infiltrate. This relationship could be explored to some extent by leveraging the observation that there are large differences between patients in the amount of unique TCR sequences detected, and also in the ‘cytotoxic T cells immune infiltrate score’ values. Is there a correlation between the # of unique TCR sequences a patient has and the value on a refined’ T cell presence score’?

Response: We thank the reviewer for the helpful and interesting suggestion. We have performed these analyses and provide them below for the reviewer’s interest. Biologically, since the number of unique clones does not directly address the number of T cells in a tumor (through clonal expansion it is possible to have hundreds of thousands of T cells with a single TCR), we do not find the results of this analyses definitively address prior concerns. However, it is nice to affirm that some relationship between the previously established immune infiltrate score correlates with increased clones. We feel that the results of the mIHC and CIBERSORT are collectively stronger data for addressing relative abundance specifically of CD8 or CD4 T cells in the tumor environment and so have not included the clone/infiltrate scores in supplement of the manuscript, as the reviewers have mentioned the manuscript is already dense with figures.

Reviewer #1 Specific Point 3

The usage of the term ‘cytotoxic T cell’ is ambiguous. Do the authors refer to all CD8 T cells? Or to only the fraction of CD8 T cells that express cytotoxicity-related genes? **Please specify.** The issue is particularly evident when the authors write about ‘exhausted CD8+ cytotoxic cell’ (Page 5, line 219). Please specify whether you refer here to all CD8 T cells that are exhausted? Or do you refer only to those (few) PD-1+ T cells that also express cytotoxic molecules? Please also specify how you define ‘exhausted’ in this context. All PD-1+ cells? Only the PD1+Tox+ ones (which are not actually cytotoxic). See e.g. Beltra et al. Immunity 2020 (John Wherry lab) and Hudson et al. Immunity 2019 (Rafi Ahmed lab).

Response: We have revised the text to more diligently and unambiguously define our employment of the word cytotoxic and exhausted, and how that relates to the existing literature.

The attributes/markers, phenotypes and response to treatment of “exhausted” cells is presently a topic of great interest and debate and is beyond the scope of this manuscript, but is reviewed here (Chung, McDonald, & Kaech, 2021). This excellent review includes and discusses the articles suggested by the reviewer. Exhaustion refers to a non-functional state of a T cell that has seen (or is currently seeing) antigen but remains unable to perform its effector functions. Analysis in human samples with unknown antigen, and in our case, the fixed state of the samples (FFPE), means we can only ascribe a “putatively” exhausted state based upon previously established gene and protein expression patterns. For both mouse and human, PD-1 and TOX-1 have prominent roles in exhaustion. As the reviewer has noted and the review cited above beautifully depicts, expression of these protein is also related to differentiation programs as well as a state of exhaustions (see review cited above). To the reviewer’s point, PD-1 and TOX-1 alone are not sufficient to ascribe the functional state of a T cell, however their presence does strongly implicate the possibility of exhaustion. To extend the gene expression data to the level of protein, we performed mIHC analysis for PD1 and TOX1 in T cells (described below in more detail). These new data are presented in figure 4h. These analyses further validate, by protein, the RNA signature of figure 4d. We have revised the manuscript to more carefully reflect the putative nature of the exhausted CD8 T cells state, and tried our best to keep our phraseology of the putative nature of this description aligned with the reserved putative label applied to similar phenotypes in existing literature. However, we acknowledge that the truest label of exhaustion is only ascribed through functional studies, which are beyond the scope of this manuscript. In addition, as the reviewer has suggested and we have discussed more in detail in Reviewer #1 Specific Point 1, we have elaborated on our use of cytotoxic and been more inclusive or specific about which immune subsets could potentially be implicated by the data.

Reviewer #1 Specific Point 4

In the same context, to aid interpretation, please specify which genes are included in the gene sets used for GSEA that are derived from papers and not readily findable in MSigDB, such as the one for ‘exhausted CD8 T cells’ that has been used. Alternatively, a clear description from which paper and what exact experimental comparison are these derived from.

Given the uncertainty regarding the genes making up the ‘exhausted CD8 T cell’ signature, it is unclear whether only exhausted CD8, or also exhausted CD4 T cells count towards the score. It is also unclear whether the enrichment for this gene set in PPBC is a consequence of that tumor type having (potentially) a larger CD8 T cell infiltrate, and therefore simply score higher because of CD8 related genes?

Response: As noted in the manuscript, we compiled gene lists (95) from several recent high profile papers concerning the tumor immune microenvironment, as well as historical manuscripts regarding postpartum biology. At the time of manuscript submission, these gene lists were not deposited in MSigDB. We have added a new supplemental resource (**Supplemental Data 01**) that provides the full gene lists with specific annotation of papers from which the individual gene lists were derived. To specifically address the reviewers important questions as to the veracity and specificity of the exhausted CD8 T cell signature in tumor samples and comparing the contribution of CD4 vs CD8 T cells to this signature we provide the following response. Like all plots generated through GSEA there is no weighted component for individual genes; rather, it’s the relative rank of members from within a composite gene profile that drive the enrichment

score and p-value compared to all other genes expressed. As such, the authors do not anticipate that the exchange (i.e. CD4 for CD8), the presence, or the absence of a singular gene alone will significantly impact a strong enrichment score like the one demonstrated in original figure 4g (new figure 4d). The gene signature was derived from a list of common genes in demonstrably exhausted CD8 T cells observed in both LCMV and tumor bearing mouse models, and was therefore labeled as “Exhausted CD8 T cell signature”. As the gene list was populated by comparing between CD8 groups (exhausted and non-exhausted), it is unlikely that any genes unique to CD8 or CD4 lineages would arise within this group, but rather only genes that are associated with immune cell states of exhaustion will be associated with this previously defined exhaustion signature. Although the signature was derived from exhausted CD8 T cells, since the signature does not depend on the CD8 designation and to avoid any implication that the signature identifies for CD8+ T cells specifically, we have removed the graph title - CD8 with the T cell exhaustion label to leave the door open for readers to consider the role of exhausted CD4 T cells. We thank the reviewer for the enhanced clarity provided in the manuscript by addressing this suggestion.

Further, we extended the inquiry of T cell exhaustion to address the impact of parity alone on this exhaustion signature. We now provided (**Supplemental figure 1b**) RNA expression data from postpartum normal breast tissue and show data depicting that, like the PPBC tissue, normal tissue is enriched for immune infiltrate score (same genes as in Figure 4a). However, unlike PPBC, normal breast tissue is not enriched for the exhausted T cell signature (compare Figure 4d with Supplemental 1c). Our goal in providing the supplemental figure 1 data is to provide a relevant comparison for evaluating the strength of the signatures we unveiled in PPBC. We have reworded the manuscript to highlight these helpful comparisons.

Reviewer #1 Specific Point 5

More relevant would be the question whether those CD8 T cells that are present in one tumor type are more exhausted than the ones in the other tumor type. This requires T cell analysis at the single-cell level: What fraction of CD8+ cells is PD-1+, what fraction is PD1+Tox+? The authors have some single-cell data from IHC, but those data are merely evaluated by comparing CD8, PD-1 and Tox staining separately, not on a per-cell basis. Furthermore, given that only the merged picture from all fluorescence channels is shown, it is not possible to judge whether the cells that stain for PD-1 also co-express CD8. With the currently presented analysis I do not consider the authors conclusion that ‘these data support PPBC being characterized by an increased presence of exhausted T cells’ warranted.

Response: The reviewer is correct in that it is possible to achieve greater clarity of putative immune functional state by performing CD45, CD3, CD8, PD-1 and Tox1 multiplex IHC staining with aligned images and cell segmentation performed to evaluate T cell and activation/exhaustion/differentiation marker expression on a per-cell basis. To address the reviewers concerns about colocalization, we have performed multiplex IHC alignment on the 16 original cases utilized in the RNAseq analysis, and have further extended these IHC analysis to include 15 cases within each cohort (n=15 PPBC, n= 15 NPBC). The mIHC images underwent co-registration, cell segmentation and image cytometry analysis utilizing FCS Express (De Novo software, CA, USA). We now provide these new results in figure 4g and accompanying

description of the results in the manuscript. Specific to the reviewers concerns a brief review of these new results is as follows. Aligned and single cell registered mIHC permitted greater specificity in evaluating the location as well as the characteristics of the T cell infiltrate. As reported previously we observed the tumor borders as the most immune cell rich regions of all the breast cancers we examined (Keren et al., 2018), providing the greatest number of events and therefore the most statistically reasoned location for examination of T cell character (Fig 4gi). As suggested by the reviewer, we provide depictions of CD4 (blue) and CD8 (purple) T cells with (Fig 4giii) and without (Fig4giv) PD and Tox1 signal overlay to provide greater perspective on the overlapping signals and therefore protein expression within these T cell compartments. Quantification of the data (Fig 4g v-vii) for the tumor border regions depict statistically increased levels of T cells in the tumor border of PPBC (Fig 4gv), with an increase most prominent for CD4 T cells. When considering the attributes of the CD4 and CD8 T cell compartments, in generally we observed statistically significant ~ 2 fold increase in PPBC of PD-1+ (black asterisk) or PD-1+Tox1+ (yellow asterisk) T cells within both the CD4 (Fig. 4gvi) and CD8 (Fig 4gvii) T cell compartments. To align the relaying of our results with the concerns raised by the reviewer of cytotoxic and exhausted T cell identity, we refrain from definitively referring to these as such T cells and instead refer to how expression of such proteins do coincide with T activation, and may contribute to the both Tfh signature observed by CIBERSORT as well as the enrichment of an exhaustion signature. In the conclusions, as we discuss the implications of our findings we extrapolate towards the hypothesis that increased expression of PD-1 on tumor border T cells may endow greater responsivity of PPBC patients to PD-1/PD-L1 checkpoint therapy. We believe moving this portion of the consideration of a putative exhaustion phenotype to the discussion assists in avoiding implications of definitive T cell character from our results.

Reviewer #1 Specific Point 6

For the IHC analysis, please specify how many fields of view are evaluated per patient (the amount of stained cells in the single presented field of view for NPBC is very low). Given that the staining pattern in the provided PPBC picture is rather clustered, please describe how the fields of view have been selected to avoid selection bias.

Response: For single channel IHC analyses, the entire stained tissue sections were scanned using an Aperio AT2 digital whole slide scanner. The scanned images were annotated by a pathologist for whole tumor area present. Since whole tumor area was included, there was no selection bias that could be introduced for these analyses. To further prevent any examiner bias, the pathologist and downstream analyst were blinded to study group and clinical data. IHC signal was quantified using a de-convolution algorithm to obtain percent positive signal. Data presented in figures 3c, 3e, and supplemental figures 2a, 2b, and 2f were all obtained from whole slide analysis and thus are not confounded by bias that can be inadvertently introduced when performing region of interest (ROI) analysis.

For CD45, CD3, CD8, PD-1 and Tox mIHC staining analysis on a per-cell basis (image cytometry), the pixel density of the scanned images necessitates region of interest (ROI) analysis, and thus the potential unintended introduction of sampling error. For these analyses ~ 3-4 ROIs were selected per case where the immune cell infiltrate was high (based on H&E and CD45 staining review). Regions with high immune cell infiltrate were selected so that a sufficient number of events needed to perform statistically justified single cell analyses for cell

character would be captured. The selection of ROIs for each case was done by 2 analysts blinded to the reproductive status of the cases, with cases randomly sorted prior to ROI selection.

Reviewer #1 Specific Point 7

While Tox is expressed by subsets of exhausted CD8 T cells, Tox expression per se cannot be interpreted as evidence for exhaustion, since also highly functional effector-memory CD8 T cells can express Tox (Sekine et al, Science Immunology 2020). In line with this, the presented IHC pictures show Tox-expressing cells that do not co-express PD-1. This should be discussed.

Response: We agree with the reviewers that with regards to the basic biology enlightened by mouse studies, and correlated in other human studies, in which PD-1 is associated with TCR activity more than merely exhaustion and also that it is not the mere expression of TOX1 but rather the relative levels of TOX1 that more specifically infer, confusingly, either a state of exhaustion (TOX1+ but low) vs effector memory differentiation (TOX1+ Hi). It would be inappropriate based upon a 4uM FFPE sections to pursue an investigation into TOX1 intensity, as the IHC intensity data between cases would not likely be reliable. Our goal was not to definitively establish an exhausted state in PPBC but rather to state the unequivocal observation that PPBC are enhanced with T cells that display protein and RNA expression profiles that others have linked to an exhausted T cell state, in more functionally supported studies, as an important point of emphasis for future clinical and basic science studies off PPBC. To the reviewer's points we have tried to clarify that our FFPE-based observations are in line with features others have associated with a functional state of exhaustion, but also emphasize the point that functional exhaustion cannot be definitively demonstrated from FFPE tissues. In addition we think it is worth emphasizing that at the time of the inception of this study, few insightful RNA signatures from FFPE were reported. To greater leverage the valuable resource of FFPE we instigated a prior study to establish and validate an FFPE RNA-Seq data acquisition and analysis pipeline (Pennock et.al, 2019). The success of this endeavor enabled the composition of this current manuscript, wherein we were able to leverage the large repository of FFPE tissues to carefully accumulate and construct a well matched PPBC and NPBC cohorts. This process required extensive chart review for parity data as well as treatment and patient characteristic data. As a result we believe we are presenting to date, the most well controlled and understood study of the molecular profile of PPBC. Compared to traditional post-menopausal breast cancer, PPBC is relatively infrequent/rare occurrence. As such, the ability to utilize retrospective chart review and patient samples from FFPE archives made this work achievable in a reasonable time scale. The trade-offs however for this advantage in composing this cohort is the limitations it places on functional studies.

Reviewer #1 Specific Point 8

Page 5, line 211 "The increased clonality of the small population caught our attention as this is unambiguously consistent with antigen-specific T cell responses 60,61 ". I do not think this statement is warranted based on neither the presented data, nor the two references. The data in figure 4f show that in PPBC, smaller clones make up a larger fraction of the repertoire than in NPBC. This does not necessarily mean that the smaller clones are also less even in size than in NPBC. Also, even if they were less even in size, this is no evidence for a more prominent (tumor) antigen-driven clonal expansion in PPBC. Clones might as well be further expanded due to inflammatory signals, and many tumor-infiltrating T cells are not actually specific for tumor

(associated) antigens, but rather for previously encountered viral infections (Scheper et al, Nature Medicine 2019 - Ton Schumacher lab)

Response: We agree with the reviewers point that the whole spectrum of clonal space can be enhanced by inflammatory environments through reactivation of memory T cells via cytokines. In previous versions of this manuscript we had described increases in small clonal space as being consistent with what is observed in most primary antigen specific T cell responses to most pathogens and vaccinations. In our attempts to simplify and condense wording of a lengthy explanation of these data, we have indeed overstated the finality of the conclusions drawn from this data by employing the word “unambiguously”. We have removed this word and reworded the remaining portion of the manuscript to appropriately soften the interpretation of these data. We also agree with what the reviewer has pointed out that most tumor infiltrating T cells are not actually specific for tumor associated antigens and therefore it is more likely that the hyper-expanded and large clone changes are irrelevant to a tumor antigen-specific response, which we believe again emphasized the importance of our observation in the expansion of the small clonal space. To this end, we have removed some of these data sets (Clonal Inequality and % Repertoire Occupancy) to the supplement (supplemental figure 2d and 2e), and reworded our findings in the results to relate rather than over-interpret these results. Further, we discuss sparingly what they could mean in the discussion and hope that our results will contribute to the growing community of TCR data, which is still establishing less equivocal metrics for inferring reinvigorated or newly primed T cell responses in human tumor tissues.

Reviewer #2, expert in breast cancer genomics (Remarks to the Author):

The authors of this manuscript performed comparative gene expression profiling of 9 cases of ER+ postpartum breast cancer (PPBC) and 7 cases of ER+ breast cancer from nulliparous (NPBC) women with the goal of identifying differences that could explain the worse outcome of PPBC. The data on outcome differences between breast cancer diagnosed in young women with differences in relation to pregnancy is not conclusive, presumably due to the relatively low number of patients in any given cohort. Understanding biological differences due to parity could potentially help the clinical management of these patients.

Numerous prior studies (reviewed in Korakiti et al. Front Onc 2020) have characterized the gene expression and/or genetic profiles of PPBC to identify biomarkers and predictors of poor outcome. P53, cell cycle-related genes (e.g., MKI67, MYC), and immune-related (e.g., PD-L1) have been shown to be differentially expressed between PPBC and non-PPBC. This current manuscript essentially confirms these prior data with novelty limited to the using only ER+ tumors and the TCR clonotype data.

Response to Reviewer #2 overall comments:

We appreciate the reviewer’s concern regarding potential overlap of our study with previously published studies designed to identify unique gene expression signatures associated with pregnancy-associated breast cancer. Specifically, the reviewer notes an important recent meta-analysis of 9 studies focused on genomic profiling of pregnancy associated breast cancer (Korakiti, Moutafi, Zografos, Dimopoulos, & Zagouri, 2020). Here, we hope to convince the reviewer that our study is a significant advancement to the field, and is not simply a reiteration of these previous studies. The primary divergence between the data we report here and prior studies

resides in the fact that previous studies are focused on breast cancers that arise during pregnancy or in close proximity to pregnancy, cancers known as pregnancy associated breast cancer (PABC) and our study focuses on postpartum breast cancer (PPBC). The studies mentioned by the reviewer focus on PABC. The classical definition of PABC is breast cancer cases diagnosed during pregnancy, with some studies extending this definition to include cases diagnosed within 6 months to one year of childbirth. Most PABC studies (including the 9 studies evaluated in the Korakiti meta-analysis) were performed under the premise that cases diagnosed during pregnancy or close to childbirth would have poor prognosis due to the exceptionally high levels of circulating estrogens and progestins during pregnancy. However, several recent publications, including a 2018 large population study by Frederic Amant, find that compared to nulliparous women, breast cancers diagnosed and treated during pregnancy do not have poorer prognosis (Amant et al., 2013; Johansson, Andersson, Hsieh, Cnattingius, & Lambe, 2011; Stensheim, Moller, van Dijk, & Fossa, 2009). Because these studies do not show a difference in outcomes between PABC and nulliparous breast cancer, our study was designed to specifically exclude PABC from analysis. Instead we focus on breast cancers that are diagnosed postpartum. Unlike a diagnosis during pregnancy, numerous outcomes studies, including a recent meta-analysis of 41 studies, repeatedly find a postpartum diagnosis to independently predict a >2-fold risk for progressing to metastatic disease (Hartman & Eslick, 2016). These poor prognosis cancers are referred to as postpartum breast cancers (PPBC). Importantly, this type of young women's breast cancer that has yet to be studied deeply at the transcriptome level.

The current literature on PABC is further confounded by the inclusion of high-risk groups (>1 year postpartum) within the study control groups. Logically, if women at higher risk, i.e., PPBC, are included as controls, the biologic impact of prior parturition will likely go unidentified. Importantly, these caveats pertain to each of the 9 papers reported in the systemic review by Korakiti et al. These papers define cases as women diagnosed during pregnancy and/or within one year postpartum, and controls as women diagnosed NOT during pregnancy or after one year of childbirth. That is, the cases include women not at increased risk of dying from their disease, and the control groups do not exclude high risk postpartum cases >1 year postpartum. Further, none of the studies reported in the Korakiti et al meta-analysis utilize RNAseq to obtain gene expression data from cases with validated parity data. Instead, most of these publications rely on quantitative PCR analysis to interrogate expression of a small, preselected set of genes. Some rely on previously published public data sets, but these publicly available data sets have limited parity data. Thus, our current study, which is focused on postpartum breast cancer with well-defined and validated parity controls, is unique. Further, we utilize whole genomic transcriptomics as the base data set. Finally, another novel attribute of our RNAseq data is that it was obtained from treatment naive breast cancer tissues, i.e. biopsies or surgical resections that occurred prior to onset of hormone, radiation or chemotherapy treatment. Thus, our data are not confounded by the presence of RNA signatures that might associate with various treatment regiments.

We agree with the reviewer that our study is small, yet we argue that aligning our study groups based on known outcomes data (i.e. excluding cases diagnosed during pregnancy because outcomes are not different) is of utmost importance if we are to advance our understanding of poor prognostic breast cancers in young women. Our study has another distinct advantage, in that the established paradigm in young women's breast cancer is that estrogen-receptor negative

disease is largely responsible for their poor outcomes, and this paper highlights the importance of continued vigilance against ER+ disease as well, both in defining the distinct molecular characteristics of PPBC compare to NPBC in young women's breast cancer disease and further extending the exploration of the consequences of these characteristics by examining outcomes in the separate compiled Young Women's Breast Cancer Cohort of 312 cases, demonstrated in Fig 5f & g. We believe the accumulation of RNA expression data for this YWBC cohort from the identified studies as well as the deposition of our clinically well described cohort of PPBC represents a significant contribution to both the PPBC and YWBC fields for future studies.

Of note, reviewer #2's recognition that the current definitions of PABC lead to confusion within the field, ultimately undermining our understanding of young women's breast cancer, contributed in part to a group of researchers within the young women's breast cancer field to submit a Commentary to *Lancet Oncology* entitled "Pregnancy-associated breast cancer definition is outdated and should not be used anymore" (Accepted, Doctopic: Analysis and Interpretation, THELANCETONCOLOGY-D-21-00418R1, S1470-2045(21)00183-2 /June issue). We thank the reviewer for highlighting this important topic, and for the opportunity to respond to this important critique. In response to reviewer comments, we have highlighted the unique aspect of our PPBC study in the revised manuscript introduction to help distinguish our work from previous published work on PABC.

The reviewer is correct in commenting that one previous study has been carried out to explore the relationship between PD-1/PD-L1 and parity status utilizing true human PPBC tissues (Tamburini et. al., 2019). In this paper, 3 cases of NPBC and 3 cases of PPBC were examined and while PD-1 expression was found to be statistically significantly elevated, it was not tied to a specific cellular identity. In our revised manuscript, we have leveraged the full mIHC platform to perform alignment and single cell segmentation in order to perform image cytometry, permitting focused quantification of PD-1 on definitive (CD45+CD3+) T cells with further breakdown by T cell compartment (Fig 4gi-vii). In addition, for mIHC cases we were able to extend these cohorts to include n=13 NPBC and n=14 PPBC cases all well annotated for the clinical features described above. We believe undertaking this effort significantly elevates the novelty of this manuscript and provides greater support to the RNA expression data relayed in Fig 4. We appreciate the reviewer pointing out the existence of this prior report which emphasizes the role of PD-L1 on lymphatic endothelial cells through employment of murine models of PPBC. We agree it represents an important conceptual advance in potential treatment of PPBC through demonstrating enhanced efficacy of checkpoint blockade in postpartum/involution mice compared to nulliparous mice. In both the original submission and resubmission we have referenced this important work and we believe our findings reported here with greater specificity and greater number of human cases, are an important contribution to bringing forward further clinical consideration, employment of checkpoint blockade specifically in PPBC.

Reviewer #2 Specific Point 1:

1. The number of cases analyzed is very few, especially considering that these are premenopausal women with differences in age, ethnicity, and tumor stage. The authors should expand their sample size and if possible, use expression data from prior publications to strengthen their conclusions.

Response: We appreciate the concern of the reviewer for conclusions drawn from a limited dataset. We would like to highlight the data in supplemental table 1b which illustrate that while there is a small but statistically significant difference in age of diagnosis between the two cohorts, all women are premenopausal, young (<45 y.o.) with ethnicity, tumor stage, and tumor subtype balanced between the RNA-Seq cohorts. Unfortunately, no other publicly available datasets are available which pair RNA expression from tumors with parity status and time since childbirth, marking the novelty of this present study. Specifically, based on the arguments presented above regarding the conflation of PABC and PPBC, it is not possible to go into previously published data sets and confidently isolate cases diagnosed in the postpartum period from those diagnosed during pregnancy, as cases are annotated as PABC, and controls do not exclude postpartum cases. Further, adding new cases and controls for prospective FFPE RNAseq would require 12-18 months of lead time-under normal operational conditions, and is not currently feasible under COVID modified operations, where at OHSU, we have only recently increased lab occupancy to 50%.

To further explore our PPBC gene expression data and help define the strength of our conclusions, we have compared our findings to expression signatures found in HEALTHY postpartum (n = 10) vs nulliparous (n = 30) breast tissues (not cancer) in **supplemental figure 1**. We also utilized the PPBC signature defined from the RNA expression profiles obtained in this study to assess its utility in predicting breast cancer prognosis more globally. Unlike any other approach in prior reports, where “parity” and “involution” signatures are composed by comparing gene expression between breast cancers in “old” and “young” patients, as well as from genes derived from rodent studies, our PPBC signature is acquired from definitive PPBC tissues. Using this human derived PPBC signature, we validated the prediction of poor prognosis through assembling a unique cohort of young women (< 45 years of age) from public data sets (n = 312). We confirm our PPBC signature predicts poor prognosis in women less than 45 years of age as well as in all and those that were clinically defined as ER+ (n = 214) analogous to the population we have interrogated in this report. These data are provided in **Figure 5f and 5g**. Aside from performing additional RNA Seq, we hope the reviewers are satisfied that we have leveraged all the available resources through their appropriate lens regarding parity data and biological subtype to corroborate our results. To date, while still small, this report represents the single largest RNA expression dataset for bonafide, treatment naïve PPBC matched to NPBC. Further we believe the extensive work undertaken in chart review and balancing of the cohorts increases the novelty and utility of this work. We hope we have highlighted this importance more clearly in the revised manuscript.

Reviewer #2 Specific Point 2:

The authors performed RNAseq on bulk FFPE tissues, thus, the observed differences can be due to differences in cellular composition, especially differences in stromal and immune cells. The authors do not show H&E images and do not mention if they controlled for cell types in the samples that were used for RNA-seq. They should at least run CYBERSORT like test to see if the predicted cell types are relatively comparable in the two groups and also perform multicolor immunofluorescence for the cell types and markers they conclude are significantly different between PPBC and NPBC cohorts in a larger set of samples.

Response: As suggested by multiple reviewers, we have performed CIBERSORT analyses to further deconvolute the bulk RNA-Seq expression signatures. Consistent with our interpretation of the gene signature results, CIBERSORT revealed CD8 T cells as representing the highest enriched immune cell population of the postpartum breast cancer group and has been included as **new Fig4 e& f** and the full results provided in **supplemental data 2**. Please see response to reviewer 1 specific point #1 for further details. In addition we have performed the mIHC to provide clarity to the differential abundance and character of immune cells (Fig 4 gi-vii) and have addressed the findings in detail in the manuscript and reviewer 1 specific point #5.

Reviewer #2 Specific Point 3:

ER+ breast cancers tend to be immune cold. Yet the authors show that based on IHC for CD45 ~10% of all cells are leukocytes (Suppl Fig. 2b), but they do not show the actual data. How was this staining quantified and how could they get such a high %? Also did this does not correlate with their prediction of CD45+ cells based on RNAseq data (Fig 4b) that shows no difference. They have to analyze the same cases using both methods and show how well they correlate. Furthermore, all IHC and IF should be quantitated based on whole slide scans.

Response: We have attempted to provide a clearer description of how the CD45 data was acquired and computed. Quantitation is based on whole slide scans and restricted to tumor area,

which was then subjected to a nuclear algorithm for ascertaining the total number of cells within the tumor. Then an AEC deconvolution algorithm was utilized to determine what fraction of associated cells were positive for AEC, which correlates to the frequency of CD45+ cells. We have provided a case by case evaluation of the results of this analytical approach with the results from RNA-Seq (left panel), which show statistical justified agreement between IHC and RNA-Seq results. As remarked by the reviewer, classically ER+ postmenopausal breast cancers are immune cold. It should be noted that *none* of these cases are classic post-menopausal breast cancer cases and instead are all breast cancers diagnosed in young women (≤ 45), so “classical” attributes of ER+ breast cancer such as “immune cold” are not necessarily expected in these samples. As mentioned previously, PPBC has yet to be evaluated extensively and the hallmarks of a “classical” case of PPBC are still being defined. Further, the average frequency of CD45 in nulliparous breast cancer in the samples we analyzed was 3.8%, which would not be considered immune hot in most contexts. However, to the reviewers point, this would not be considered immune excluded, either. When we expanded this cohort to a total of 30 cases for NPBC ($n = 15$) and PPBC ($n = 15$), the NPBC average rose to 5.2% (now depicted in **supplemental figure 2b**). Current regards to the descriptions of immunologically hot vs cold tumors more commonly refers to the level, placement and composition of T cell infiltrate more

than the presence of CD45 cells (Galon & Bruni, 2019). In some cases (such as melanoma) it has been demonstrated that the most immunologically hot tumors are comprised of as much as 20-60% of total nuclei being derived from T cells alone. By these metrics all of the cases examined above are relatively immunologically cold. Furthermore, in the breast specifically we don't expect a completely devoid presence of immune cells as immune cells have repeatedly been demonstrated to be an integrated component of normal breast microenvironment in both mice and humans (Betts et al., 2018; Gouon-Evans, Rothenberg, & Pollard, 2000; Jindal, Narasimhan, Borges, & Schedin, 2020; Van Nguyen & Pollard, 2002).

Reviewer #2 Specific Point 4:

The TCR clonotype repertoire data is very confusingly described and presented. How can PPBC have ~4 times more unique TCRs (Fig 4c), but still have the same clonotype diversity (Fig 4d) and at the same time have ~2x higher clonality index (Fig 4e)? the authors have to describe and present these data in more clearly.

Response: We appreciate the reviewer voicing their concern over the presentation of the TCR data. We have reworded this result section to be more intuitive and informational. To assist in providing clarity we have moved the implications of the results to discussion. To aid in clarity and interpretation we have removed the inverse related TCR metric of normalized entropy altogether, moved the clonal space analysis to **supplemental fig. 2e** and included the Gini Index in **supplemental Fig 2d**. We hope our new wording and presentation of the data aids in the readability and understanding of the data.

Reviewer #2 Specific Point 5:

The differences observed in TCR clonotype diversity could be due to differences in overall peripheral blood TCR diversity. The authors should analyze peripheral blood to determine this.

Response: The reviewer is correct that the diversity of TCR sequences in the tumor may very well reflect the peripheral blood TCR diversity. Unfortunately, due to the archival nature of the FFPE samples which were collected historically over broad range of time we are not able to analyze the matched peripheral blood. Furthermore, it would be anticipated that analysis of the peripheral blood would only aid in distinguishing the clones that exist as part of the tissue resident memory T cell population (Trm). As noted by reviewer 1 (the t-cell expert) and discussed in response to reviewer 1, Trm have been described to have a greater clonality (reduce TCR diversity), respond to inflammatory stimulus, and *are* more likely to be present in the large and hyperexpanded clonal distributions. As also remarked by reviewer 1, these Trms are not popularly ascribed to be specific to tumor antigens. Nonetheless, their contribution to the anti-tumor response is still being assessed as bystander responders to inflammation generated from both the tumor and in response to therapy. Approaches to hone in on tumor specific T cell subsets through subtraction of clones present in peripheral blood ignore the fact that, to the best of our knowledge, the majority of T cell priming occurs in the tumor draining lymph node or other secondary lymphoid structures in both mouse and humans, and therefore necessitates recirculation in the peripheral blood stream before extravasation and arrival at the tumor site. Likewise, copious data from antigen-specific T cell responses to vaccinations in mice, humans and non-human primates clearly demonstrate that the blood is one of the early and most sensitive sites to detect antigen driven specific T cell expansion. Given this, the most informative use of

peripheral blood for TCR repertoire analysis must be paired with pre and post insult conditions. As is the case for most archival tumor studies, we have limited to no access to blood samples in general, let alone from these subjects prior to a tumor diagnosis. Further comments regarding TCR analysis are addressed in response reviewer 1 comment 7. We hope the changes we have made to the text regarding our TCR analysis will provide a clearer read.

Minor comment:

1. The manuscript text is very dense (both space and text) and not easy to read. It would be great to improve this.

Response: Thank you for this important feedback, we have edited the manuscript with this critique in mind.

2. It would be helpful if the authors indicate the figure number on the figures.

Response: We are adhering to the guidelines of formatting for figure submissions provided by Nature Communications. If accepted, figures will be associated with a number figure legend. We apologize for any confusion instigated by the manuscript distribution software for reviewers.

3. “Unique numbers of T-cell receptor repertoire (TCR)” in legend of Figure 4 makes no sense. The authors presumably meant Unique numbers of TCRs (TCR stands for T cell receptor)?

Response: Thank you for catching this error! We have corrected this figure.

Reviewer #3, expert in multiplex immunohistochemistry and biomarkers (Remarks to the Author):

Jindal, Pennock and colleagues present a study characterizing the features of breast cancer arising in young women with a focus on the role of parity in disease pathogenesis (NCOMMS-20-41597). The area of study is very important, the manuscript is well written and the claims are very interesting however one major challenge with this study is the small number of cases that were accessible for analysis (9 PPBC, 7 NPBC) and the molecular heterogeneity of the 9 PPBC of which 4 had TP53 mutation. Despite the authors’ significant effort to match samples between the cohorts, it is difficult to assess how strong the conclusions are about breast cancers that arise in this particular clinical setting and whether follow on steps such as generating a so called ‘PPBC signature’ are appropriate in the context of a small and heterogeneous cohort.

Response to Reviewer #3 overall comments: We appreciate the reviewer’s recognition of the significant efforts to create an informative cohort for RNA-Seq expression profiling in the unique case of young women’s breast cancer from the limited molecular data that is available from chart review prior to the onset and acquisition of the expression data. The name of the ‘PPBC signature’ was not intended to reflect an exhaustive and definitive gene expression pattern for all PPBC cases, but instead to reflect the key comparison in the patient population from which the signature was derived in this paper. While the reviewer is concerned about the molecular heterogeneity of the PPBC population, it should be remarked that computationally,

Single Sample TP53 Regulon Score

heterogeneity should be nullified not magnified when deriving a signature. This is demonstrated to be true for this figure set in figure 5d where use of the PAM50 subtype genes fails to delineate samples based upon their computed PAM50 subtype, nor parity status, indicating that the gene expression profiles of the PAM50 gene set are too diverse to contribute to the establishment of a signature. It is the elements that are most common through all of the PPBC samples that should rise to the surface with our analyses. While it is true that half of the PPBC samples have previously documented and P53 mutations, the remaining samples also have significantly altered P53 regulon scores,

implicating p53 biology in PPBC in the absence of mutation (shown by single sample regulon score). Further in line with the reviewers concern regarding TP53 we have enable the tracking of the ICGC P53 mutants throughout the manuscript with the use of orange filled circles. This does not preclude the possibility that other P53 mutants may exist in both the PPBC and NPBC cases that are not identified by our mutant identification approach. By tracking of these TP53 mutants throughout the manuscript it is clear that is not the identified TP53 mutants behaving as outliers that are driving the gene expression profiles we are emphasizing. We hope that with tracking of these TP53 mutants throughout the manuscript and the intriguing survival results derived from the employment of the PPBC expression profiles in which PPBC signature Hi young women breast cancer patients have poorer outcomes (Fig 5f, n=312) even in the case of ER+ disease (Fig 5g, n=214) emphasis is drawn to the potential importance of these initial findings and worthiness of our endeavors to characterize the molecular nature common in ER+ PPBC compare to other young women’s nulliparous breast cancer.

Reviewer #3 Specific Point 1:

It is unclear if some of the observed findings are driven by the preponderance of P53 mutant cases in the PPBC samples that were available. It would ultimately be helpful to compare in the gene set enrichment analyses how P53 wild type PPBC compare with P53-mutant PPBC and with NPBC? It would help in the analyses and plots to indicate which samples are p53 mutant (colouring of dots in the dot plots is a start – for instance in Fig 3b, c and 3e). Were the authors able to assess if any of these patients had a germline mutation of p53 (Li Fraumeni); which samples were P53 mutant and what were the mutations (should be listed in supplementary table 1). Are P53 mutations enriched in PPBC (please see point further below on the use of other samples that were not included in the RNA analysis).

Response: To address reviewer's request to compare gene set enrichment analyses on how P53 wild type PPBC compare with P53-mutant PPBC and with NPBC, we have color coded the cases in figure 3e and throughout the manuscript based upon the determined TP53 mutational status with orange dots. We observe that p53 mutation cases had no effect on driving cell cycle, proliferation, immune infiltration score, TCR reads, clonality index or CIBERSORT data for CD8 cells, or ESR1 regulon scores for the PPBC cases when compared to NPBC, thus p53 mutation alone does not solely account for the biological differences we observe between NPBC and PPBC cases. Since we only have FFPE tissues for the tumors, we cannot access the matched normal tissue to check the germline mutation of p53. But we are confident of our detected TP53 mutations because the exact mutated locus are also reported in TCGA samples.

Reviewer #3 Specific Point 2:

One set of questions relates to how PPBC and NPBC compare with ER+ breast cancers in the general population and whether some of the gene set categories (such as mammary gland involution) that are enriched in PPBC (vs. NPBC) would be enriched in ER+ breast cancer in the general (and older) population vs NPBC. It would help the reader if the basis for how this signature was generated were explained in the text and not solely referenced.

Response: We agree with the reviewer's suggestion that understanding the gene set signature in the ER+ PPBC and NPBC among younger and older women is important. Our ER gene signature was generated entirely from younger women, and whether this signature might persist in older parous, but not nulliparous women remains to be determined. The barrier to successfully addressing this question is the lack of reliable data in current public data bases related to time between last pregnancy and breast cancer diagnoses. Further, in order to control the confounding factors and to focus on PPBC vs NPBC, our own sample size is small and not practically big enough to address this question in the general population, thus it remains beyond the scope of the current manuscript. In the methods, figure legends and supplemental materials provided, there is sufficient detail for other bioinformatics/computational biology enabled investigators to replicate and employ the PPBC signature to their future studies. As part of data sharing, open access compliance, data integrity and collegiality, the authors respond with assistance to any reasonable requests for methodological assistance.

Reviewer #3 Specific Point 3:

The reliance on the scoring of ER IHC from clinical pathology reports seems suboptimal in this study because the information plays an important role in the authors interpretation of their findings. The quantification of the frequency of ER positive tumor cells and perhaps even the level of ER protein expression on a single cell level may contain meaningful information and that information is accessible to the authors using their multiplexed IHC platform. In fact, some of the clinical cases were only scored as positive or negative without providing a percentage - were those 5-10% positive or 100% positive. What is the threshold at the authors' institution? A more careful characterization and assessment of the expression of ER seems therefore advisable.

Response: We agree with the reviewer that our reliance on clinical chart extraction to obtain frequency of ER positive tumor cells is not optimal given how central ER positivity is to our data analyses. As recommended, we have gone back and stained these cases using the clinically validated ER antibody (Agilent/Dako, CA). Staining, data capture and analysis were all

performed with the operator blinded to study group. ER expression levels were independently assessed by a pathologist (and corroborated) by a second operator. Firstly, we confirm the original chart review data, plus were able to provide % ER positive data for the three cases where that detailed ER data were missing. With a single NPBC case exception, all cases had ER positivity confirmed to be $\geq 40\%$, with no differences observed between nulliparous and PPBC cases. Overall the average ER expression was $\sim 75\%$. Further, as described above, we expanded our data set to include 14 additional cases. We do not find differences in % ER positivity between nulliparous and PPBC cases in the original cases utilized for RNAseq, nor in the expanded data set (new **Supplemental figure 3a and 3b**). We also assessed the intensity of ER staining and find no statistically significant differences in ER stain intensity between NPBC and PPBC groups (new **Supplemental figure 3b**).

To reflect these new ER data, we have replaced the previous % ER positive data results obtained from chart review and presented in original Table 1 with pathological calls of ER positivity based on our own staining results and have added a new supplemental figure to the manuscript (new **Supplemental figure 3a and new Supplemental table 1a**).

Reviewer #3 Specific Point 4:

The authors screened 40 cases but only 16 were deemed useful for RNA studies. However, very likely some of the cases would have been suitable for tissue imaging (P53 IHC to further assess the frequency of P53 mutation in the PPBC vs NPBC groups; ER IHC to see if the levels are the same between the two groups). Having added such cases would have allowed the authors to expand the size of the cohort for some analyses and could have in part helped strengthen some of the conclusions.

While the 8-plex tissue imaging as used in this study does provide an orthogonal method for assessing a few of the observations made using gene set analyses of RNA data, there does seem to be a missed opportunity to characterize more carefully these uncommon samples for several reasons.

Response: We are thankful to the reviewer for mentioning the idea of performing multiplex staining of additional cases that were inadequate to proceed for the RNA sequencing. Since we received these comments, and as evidence from our responses above, we have gone back and stained all cases with available tissue for multiplex IHC. Our multiplex IHC now include additional 14 cases and results in a total sample size of 30 cases comprising PPBC (n = 15) and NPBC (n = 15) cases. The new robust data is included in the revised figures 3c,3e, 4g (v-vii), supplemental Fig. 2b,2f ,3a and 3b. As before, most of the significant differences observed in the smaller cohort are seen in the larger cohort. Further, while the IHC data comparison for different markers for the original 16 and now 30 cases for the PPBC and NPBC are

represented throughout our rebuttal and manuscript, we also provide here a snapshot of some metrics assessed as single analytes as whole slide analyses for ease of clarity for reviewer #3. Some of these data have been replaced in the main figures in favor of image cytometry data, providing greater detail with regard to cellular identity in response to other reviewer suggestions (Fig 4g).

Reviewer #3 Specific Point 5:

A tumor marker (e.g., keratin) is not included in the panel and therefore the ability to interpret ‘tumor cell proliferation’ is fraught because the quantification of Ki67 likely includes immune cells and stromal cells which can often be found to represent a fairly large fraction of proliferative cells in samples. Elevated p53 could in principle be used as a tumor marker in the four TP53 mutant cases but the remaining cases have no tumor marker. It is important to note that Ki67 is known to have graded expression (Miller et al., 2018) so setting gates for determining Ki67+ and Ki67- cells is not straightforward (please see next comment).

Response: We agree with the reviewer that measuring Ki67 positivity within the tumor is representative of both proliferating tumor and immune cells. Further, as the reviewer points out, we do not have a tumor specific biomarker that can be applied to all cases. We have attempted to address the reviewer’s concerns as follows. Firstly, for the Ki67 data presented in our original submission (Fig 3c) the Ki67 data were obtained from the entire tumor area present on the sections, thus potential for area-selection bias is not relevant to this data set. Specifically, we digitally scanned the entire Ki67 stained slides using an Aperio scanscope. The scanned images were then annotated for all the tumor area that was present on each slide and a semi-quantitative algorithm was run to get the % positive proliferating cells. Any cell that was positive for nuclear Ki67 (irrespective of the gradient) was considered as proliferative signal. As the reviewer has noted, Ki67 does provide a gradient of staining. In the field there is much variability in how to interpret and analyze staining Ki67 which has limited the broad clinical use of Ki67 stains in treatment decisions. We have quantified our Ki67 staining in line with recommendations based upon extensive meta-review and consortium recommendations (Inwald et. al., 2013). Since our last submission we have added 14 additional cases to the dataset and see a similar trend between groups, i.e., there is not a significant difference in overall Ki67 stain between nulliparous and postpartum breast cancer cases (**new figure 3c**). Secondly, as the reviewer points out, this Aperio analysis does not differentiate the proliferating tumor and immune cells. To better address this question, we performed a negative selection analysis where we captured intra-tumoral CD45 negative cells in our image cytometry as a better representation of tumor cells. Consistent with our Aperio based IHC analysis, we find no difference in putative tumor cell proliferation between nulliparous and postpartum cases (graph shown adjacent) This second analysis was performed by aligning three images in FCS

Express, the hematoxylin stain, and CD45 and Ki67 IHC images. With FCS express analysis, the size of any one area of tissue that can be analyzed is limited to 60,000 events, which may result in operator selection bias. Below we discuss how tissue areas for FCS express analysis were selected, which hopefully the reviewer will agree were performed with an emphasis on minimizing the potential for operator bias.

Reviewer #3 Specific Point 6:

The approach for analysis is not well described. Restricting analyses to regions of interest seems unwarranted and in addition it does not seem that important information about the analysis was provided in the manuscript – based on what criteria were the regions selected (randomly? informed by some feature?, restricted to the tumor compartment or the tumor stromal interface?), how many regions were quantified per case, how many cells were analyzed, how do the authors know that these are representative fields and that their results would not be different if other ROIs were selected? If whole slide imaging was performed, whole slide analysis seems to be the preferred approach and would allow for a more sophisticated characterization of these samples. How did the authors go about calling positive and negative cells particularly in cases like Ki67 where expression was likely not bimodal but rather continuous/graded? The use of ROIs for analysis seems important in light of data like the trend toward a significant increase in the CD45 population in the PPBC but the difference is deemed non-significant whereas the levels of CD8 are significant. Is this a regional or global effect?

Response:

All of the original quantitative IHC data provided in our original manuscript were collected from whole tissue sections, and thus the data are not confounded by potential ROI bias. We have clarified the methods section to reflect this. We have also clarified the method section to describe how Ki67 and ER signals were captured, as these two biomarkers can vary based on % positivity and intensity.

For the newly added single cell CD45, CD3, CD8, PD-1 and Tox-1 mIHC staining analysis, the pixel density of the scanned images is an inherent limitation of FCS express image analysis software, the single cell mIHC analysis technology that integrates into our mIHC analysis pipeline. The solution is to utilize ROIs for analysis. We focused on the tumor border area as most of the tumor infiltrating lymphocytes are aggregated at tumor border. For these analyses ~ 3-4 ROIs were selected per case where the immune cell infiltrate was high (based on H&E and CD45 staining review). Regions with high immune cell infiltrate were selected so that a sufficient number of events (number of single CD3+ cells) needed to perform 5 channel single cell analyses would be captured. The selection of ROIs for each case was done by 2 analysts blinded to the reproductive status of the cases, with cases randomly sorted prior to ROI selection. These ROI were then processed and analyzed using the mIHC pipeline listed in the methods section. The data was plotted as %CD45 cells (new figure 4g).

Reviewer #3 Specific Point 7: A richer set of markers (keratin and ER mentioned above for instance) would have been very useful in this study. The authors have reported the role of COX2 in YWBC but that was not explored. The levels of Tregs was not monitored. Exhaustion is a complex phenotype to characterize on the level of marker expression so the use of additional exhaustion markers such as LAG3 and TIM3 would have been very informative as would activation markers such as granzyme B and perforin (and Ki67). Those markers would allow a more careful evaluation of what appears to be conflicting results from the gene set enrichment analysis – T cell presence and activation in Figure 2c and exhaustion of CD8 T cells in Figure 4g. PD1 on its own is not a marker of exhaustion and can be present during T cell activation (along with Ki67) – did the authors score double positives for PD1 and TOX? Also, the spatial

arrangement of the activated and dysfunctional immune cells is not presented - where are these populations relative to the tumor cells ... are they in the tumor compartment, at the interface or at a distance from the tumor border. Are the levels of dysfunctional T cells influenced by p53 status of the PPBC? In an ideal setting the authors would provide their IHC data to the public for analysis, particularly because these are such uncommon samples and therefore of significant public interest.

Response - We agree with the reviewer that the list of putative activation and exhaustion markers that can be used to phenotype T cells is rapidly expanding, and that our limited selection of markers is not definitive. Since we have not validated LAG3, TIM3, granzyme B or perforin for multiplex analysis, we felt we could not accomplish these analyses within the time frame of a resubmission. A detailed response regarding PD-1 and TOX-1 are provided in response to (t cell expert) reviewer 1’s similar concerns. Please see reviewer 1 remarks 3,4,& 6 and our accompanying response and amendments to the paper. From reviewer 1’s review we would like to emphasize their statement, “The presented data are - as the authors state - consistent with the hypothesis that PPBC contain higher numbers of tumor-infiltrating CD8 T cells, and that these T cells are exhausted and more clonally expanded. However, alternative hypotheses are not investigated, and formal proof for the hypothesis is not

provided.” In addition we provide here per the reviewers request the location of the identified TP53 mutants in the newly provided T cell TOX1 and PD-1 from image cytometry (new Fig 4g). Interestingly, there is no obvious correlation of the location of these mutants and the character of T cells with regard to PD-1 or Tox-1 expression. In addition to our GEO deposition of the RNA-Seq data, we have included the entire data set for all plots in this paper as supplemental data 03, and data is provide by case with annotations of TP53 mutation and parity status. This is provided that reviewers and other interested investigators can pursue additional hypotheses which can not be addressed within the scope and space of the present manuscript. We hope that we have altered language appropriately and provided additional data to satisfy both reviewer 1 and reviewer 3 to an acceptable degree on this common issue of the role and identity of T cells in PPBC.

Reviewer #3 Specific Point 8:

The authors can be more careful with their descriptions – for instance they describe in the discussion a “pronounced T cell presence” ... but the analysis is only for CD8 T cells and does not include CD4 cells for instance.

Response – We appreciate the careful review of our words regarding our description of T cell responses and we have carefully reviewed and amended our language accordingly. In addition as has been mentioned above in response to reviewer 1 our new analyses also includes consideration of CD4 T cells as well.

Reviewer #3 Additional points/questions:

Presumably “whole genome profiling” of the breast tissue from healthy patients is RNA-seq? Why did the PPBC cohort not receive hormone therapy? Is that unexpected?

Response: Thank you for drawing our attention to this phrase. We have inserted “whole transcriptome profiling” to correct and provide greater clarity to the methodology employed regarding RNA expression profiling of healthy tissues. With regard to treatment, a strength of our FFPE RNAseq data is that it was obtained from treatment naïve breast cancer tissues, obtained from biopsies or surgical resections that occurred prior to onset of hormone, radiation or chemotherapy treatment. Thus, our data are not confounded by the presence of RNA signatures that might associate with various treatment regimens.

References:

- Amant, F., von Minckwitz, G., Han, S. N., Bontenbal, M., Ring, A. E., Giermek, J., . . . Loibl, S. (2013). Prognosis of women with primary breast cancer diagnosed during pregnancy: results from an international collaborative study. *J Clin Oncol*, *31*(20), 2532-2539. doi:10.1200/JCO.2012.45.6335
- Betts, C. B., Pennock, N. D., Caruso, B. P., Ruffell, B., Borges, V. F., & Schedin, P. (2018). Mucosal Immunity in the Female Murine Mammary Gland. *J Immunol*, *201*(2), 734-746. doi:10.4049/jimmunol.1800023
- Chung, H. K., McDonald, B., & Kaech, S. M. (2021). The architectural design of CD8+ T cell responses in acute and chronic infection: Parallel structures with divergent fates. *J Exp Med*, *218*(4). doi:10.1084/jem.20201730
- Davoli, T., Uno, H., Wooten, E. C., & Elledge, S. J. (2017). Tumor aneuploidy correlates with markers of immune evasion and with reduced response to immunotherapy. *Science*, *355*(6322). doi:10.1126/science.aaf8399
- Galon, J., & Bruni, D. (2019). Approaches to treat immune hot, altered and cold tumours with combination immunotherapies. *Nat Rev Drug Discov*, *18*(3), 197-218. doi:10.1038/s41573-018-0007-y
- Gouon-Evans, V., Rothenberg, M. E., & Pollard, J. W. (2000). Postnatal mammary gland development requires macrophages and eosinophils. *Development*, *127*(11), 2269-2282. Retrieved from <https://www.ncbi.nlm.nih.gov/pubmed/10804170>
- Hartman, E. K., & Eslick, G. D. (2016). The prognosis of women diagnosed with breast cancer before, during and after pregnancy: a meta-analysis. *Breast Cancer Res Treat*, *160*(2), 347-360. doi:10.1007/s10549-016-3989-3
- Inwald, E. C., Klinkhammer-Schalke, M., Hofstädter, F., Zeman, F., Koller, M., Gerstenhauer, M., & Ortmann, O. (2013). Ki-67 is a prognostic parameter in breast cancer patients: results of a large population-based cohort of a cancer registry. *Breast cancer research and treatment*, *139*(2), 539–552. <https://doi.org/10.1007/s10549-013-2560-8>
- Jindal, S., Narasimhan, J., Borges, V. F., & Schedin, P. (2020). Characterization of weaning-induced breast involution in women: implications for young women's breast cancer. *NPJ Breast Cancer*, *6*, 55. doi:10.1038/s41523-020-00196-3
- Johansson, A. L., Andersson, T. M., Hsieh, C. C., Cnattingius, S., & Lambe, M. (2011). Increased mortality in women with breast cancer detected during pregnancy and different

- periods postpartum. *Cancer Epidemiol Biomarkers Prev*, 20(9), 1865-1872.
doi:10.1158/1055-9965.EPI-11-0515
- Keren, L., Bosse, M., Marquez, D., Angoshtari, R., Jain, S., Varma, S., . . . Angelo, M. (2018). A Structured Tumor-Immune Microenvironment in Triple Negative Breast Cancer Revealed by Multiplexed Ion Beam Imaging. *Cell*, 174(6), 1373-1387 e1319.
doi:10.1016/j.cell.2018.08.039
- Korakiti, A. M., Moutafi, M., Zografos, E., Dimopoulos, M. A., & Zagouri, F. (2020). The Genomic Profile of Pregnancy-Associated Breast Cancer: A Systematic Review. *Front Oncol*, 10, 1773. doi:10.3389/fonc.2020.01773
- Miller, I., Min, M., Yang, C., Tian, C., Gookin, S., Carter, D., & Spencer, S. L. (2018). Ki67 is a Graded Rather than a Binary Marker of Proliferation versus Quiescence. *Cell Rep*, 24(5), 1105-1112 e1105. doi:10.1016/j.celrep.2018.06.110
- Pennock ND, Jindal S, Horton W, Sun D, Narasimhan J, Carbone L, Fei SS, Searles R, Harrington CA, Burchard J, Weinmann S, Schedin P, Xia Z. (2019) RNA-seq from archival FFPE breast cancer samples: molecular pathway fidelity and novel discovery. *BMC Med Genomics*. 2019 Dec 19;12(1):195. doi: 10.1186/s12920-019-0643-z. PMID: 31856832; PMCID: PMC6924022.
- Stensheim, H., Moller, B., van Dijk, T., & Fossa, S. D. (2009). Cause-specific survival for women diagnosed with cancer during pregnancy or lactation: a registry-based cohort study. *J Clin Oncol*, 27(1), 45-51. doi:10.1200/JCO.2008.17.4110
- Tamburini, B., Elder, A. M., Finlon, J. M., Winter, A. B., Wessells, V. M., Borges, V. F., & Lyons, T. R. (2019). PD-1 Blockade During Post-partum Involution Reactivates the Anti-tumor Response and Reduces Lymphatic Vessel Density. *Frontiers in immunology*, 10, 1313. <https://doi.org/10.3389/fimmu.2019.01313>
- Van Nguyen, A., & Pollard, J. W. (2002). Colony stimulating factor-1 is required to recruit macrophages into the mammary gland to facilitate mammary ductal outgrowth. *Dev Biol*, 247(1), 11-25. doi:10.1006/dbio.2002.0669

Reviewers' Comments:

Reviewer #1:

Remarks to the Author:

The manuscript is very much improved, and all my concerns have been adequately addressed.

One minor note: In Figure 4vi and 4vii it is unclear what comparison the yellow and the black star relate to.

Reviewer #2:

Remarks to the Author:

The authors have responded to each of the reviewers' specific comments and made some revisions to the manuscript, however, they did not really perform additional experiments to address these.

Importantly, the number of samples analyzed remains fairly low.

CIBERSORT is not really accurate to dissect immune subtypes – even if it's widely used, but it is very inaccurate – especially for certain subsets of cells. The authors should have performed multiplex immunofluorescence to better characterize the immune environment of the parity-associated and subtype & age matched control tumors.

Reviewer #3:

Remarks to the Author:

In the revised manuscript "In young women's breast cancer parity status associated with expression signatures of T cell activation, cell cycle and reduced ER signaling" the authors have adequately responded to prior specific points. Providing multiplexed IHC data on the additional samples is helpful as is the ER IHC quantification on the samples to validate clinical scoring.

One point that remains unclear is how the authors can align IHC images of CD45 and Ki67 from presumably serial sections to enable single cell measurements (these are not entirely the same cells, particularly lymphocytes which are smaller than tumor cells and change substantially between 4-5 micron sections) ... and is the %CD45- Ki67+ cells just shown in the reviewer response and not the main manuscript? This is not really addressing the question, and therefore this question should be approached in future work.

Also, the ROI based analysis for multiplexed imaging, while understandable, should also be surmounted in the future.

As the authors note in their response, there are a number of limitations to the study; these might best be presented in paragraph dedicated to limitations in the discussion; this would help the reader.

One point of note from the authors' findings and mentioned in the fifth paragraph of the discussion: there are studies in tissue culture cells (PMID: 30067968) and more recently in human tissues (<https://doi.org/10.1101/2021.05.16.443704>) that show that Ki-67 is a marker of the G2 phase of the cell cycle (missing G1) and that Ki67 used as a single marker is not sufficient to capture cell proliferation phenotypes and dynamics that may be observed in RNA sequencing space, possibly partly explaining the conundrum the authors mention.

RESPONSE TO REVIEWERS' COMMENTS

Reviewer #1 (Remarks to the Author):

The manuscript is very much improved, and all my concerns have been adequately addressed.

One minor note: In Figure 4vi and 4vii it is unclear what comparison the yellow and the black star relate to.

Author Response: The authors have modified the results and figure legends to more clearly designate what the statistical stars refer to.

Reviewer #2 (Remarks to the Author):

The authors have responded to each of the reviewers' specific comments and made some revisions to the manuscript, however, they did not really perform additional experiments to address these. Importantly, the number of samples analyzed remains fairly low.

Author Response: We agree with the reviewer that our study size is not large enough to be fully representative of the myriad of interaction between young women's breast cancer (YWBC) and parity history. We remind the reviewer that YWBC is relatively rare compared to postmenopausal breast cancer, and that parity data are poorly annotated or lacking in clinical data sets resulting in most cases being insufficiently annotated for inclusion in our study. Our requisite for treatment-naïve specimens further restricted the number of cases that could be incorporated into our study. In response to this previous critique, we performed several new IHC stains and analyses on both the existing cases, and almost doubled the size of the cohorts ($n = 15$ for each) for certain analyses. In addition to the existing RNA Seq data, new analyses were conducted as requested (CIBERSORT) and those results were further followed up on with single cell multiplex IHC analyses. With these inherent limitations in mind, we hope the reviewer agrees with us that this expanded cohort size is sufficient to draw the conclusions presented in the manuscript. Our long-term goal is to continue to build on our well-annotated YWBC tissue cohorts, and expand the molecular profiling of postpartum breast cancers. However, such tissue expansion is outside the scope of this current submission.

Reviewer #2 (Remarks to the Author):

CIBERSORT is not really accurate to dissect immune subtypes – even if it's widely used, but it is very inaccurate – especially for certain subsets of cells.

Author Response: The authors agree with the reviewer and have included a statement in results regarding the limitations of CIBERSORT and provide that statement as a rationale for performing IHC to assist in resolving the ambiguity from the CIBERSORT data (page 6, line 251-52).

Reviewer #2 (Remarks to the Author):

The authors should have performed multiplex immunofluorescence to better characterize the immune environment of the parity-associated and subtype & age matched control tumors.

Author Response: CIBERSORT was performed at the request of previous reviewers. In addition, the authors performed and presented the results of multiplex IHC in the revised manuscript in figures 4g i-vii, which reasonably supported the conclusions from the CIBERSORT analysis (within the bounds of its limitations) as discussed by the authors. The authors agree that a more extensive characterization of the

immune microenvironment beyond that implicated in RNA Seq expression profiling is an intriguing potential direction, however the authors contend that this is beyond the scope of the current manuscript (which presently contains 5 figures and 3 supplemental figures).

Reviewer #3 (Remarks to the Author):

In the revised manuscript “In young women’s breast cancer parity status associated with expression signatures of T cell activation, cell cycle and reduced ER signaling” the authors have adequately responded to prior specific points. Providing multiplexed IHC data on the additional samples is helpful as is the ER IHC quantification on the samples to validate clinical scoring.

One point that remains unclear is how the authors can align IHC images of CD45 and Ki67 from presumably serial sections to enable single cell measurements (these are not entirely the same cells, particularly lymphocytes which are smaller than tumor cells and change substantially between 4-5 micron sections) ... and is the %CD45- Ki67+ cells just shown in the reviewer response and not the main manuscript? This is not really addressing the question, and therefore this question should be approached in future work.

Author Response: The authors apologize for the ambiguity in relaying the methodologies for IHC staining. As indicated in the manuscript, the authors have conducted a corresponding multiplex IHC assay. This assay is a cyclical process of staining for various analytes on the same section of tissue, not the overlaying of results from serial sections, which we agree with the reviewer would complicate interpretation. This confusion may come from our presentation of data both by 1) traditional IHC single analyte and % positive tumor cells approach and 2) employment of single cell deconvolution using serial multiplex IHC images obtained from performing mIHC on a single section. Using mIHC, we can delineate Ki67 in CD45 positive cells with high confidence.

The reviewer is also correct that we have not included the multiplex single cell CD45⁺Ki-67⁺ data that we had provided in our response to reviewers during the last revision cycle. We made this decision based on the fact that this Ki67 IHC data supported the results we present in the main manuscript, does not alter our interpretation of that data, and would add unnecessary length to the manuscript, which is already dense.

Reviewer #3 (Remarks to the Author):

Also, the ROI based analysis for multiplexed imaging, while understandable, should also be surmounted in the future.

Author Response: The authors agree that the computational restrictions that currently force the ROI-based approach for multiplex IHC are a regrettable reality and share the reviewer’s enthusiasm for the day when such bottlenecks do not restrict analysis. We point to the details relayed in our methods which described the acquisition of multiple ROIs for each sample from discrete tumor locations (tumor border and intratumoral) as a strength to our approach to assess the evident heterogeneity in the tumor microenvironment. The single dot presented in the data represents the average results from multiple ROIs per sample from a given location type (intratumoral or tumor border).

Reviewer #3 (Remarks to the Author):

As the authors note in their response, there are a number of limitations to the study; these might best be presented in paragraph dedicated to limitations in the discussion; this would help the reader.

One point of note from the authors' findings and mentioned in the fifth paragraph of the discussion: there are studies in tissue culture cells (PMID: 30067968) and more recently in human tissues (<https://doi.org/10.1101/2021.05.16.443704>) that show that Ki-67 is a marker of the G2 phase of the cell cycle (missing G1) and that Ki67 used as a single marker is not sufficient to capture cell proliferation phenotypes and dynamics that may be observed in RNA sequencing space, possibly partly explaining the conundrum the authors mention.

Author Response: We thank the reviewer for their assistance in relaying the limitations of the Ki67 in assessing the proliferation state of a cell. We have included this limitation and relevant references in the body of the paper. In addition, as recommended by the reviewer, we have provided an additional statement of the chief limitations of our study in the discussion, which are the limited sample size and the sole employment of FFPE tissues for obtaining biological insight.